# Modeling of Closure of Metallurgical Discontinuities in the Process of Forging Zirconium Alloy

**DOI:** 10.3390/ma16155431

**Published:** 2023-08-02

**Authors:** Grzegorz Banaszek, Kirill Ozhmegov, Anna Kawałek, Sylwester Sawicki, Alexandr Arbuz, Abdrakhman Naizabekov

**Affiliations:** 1Metal Forming Department, Częstochowa University of Technology, ul. J.H. Dąbrowskiego 69, 42-201 Częstochowa, Poland; grzegorz.banaszek@pcz.pl (G.B.); kvozhmegov@wp.pl (K.O.); kawalek.anna@pcz.pl (A.K.); 2Mechanical Engineering Department, AbylkasSaginov Karaganda Technical University, 56 Nursultan Nazarbayev Ave., Karaganda 100027, Kazakhstan; mr.medet@outlook.com; 3Core Facilities Department, Nazarbayev University, 53 KabanbayBatyr Ave, Astana 010000, Kazakhstan; 4Rudny Industrial Institute, 50Let Oktyabrya Street 38, Rudny 111500, Kazakhstan; naizabekov57@mail.ru

**Keywords:** closurefoundry voids, zirconium ally, Forge, numerical modeling, forging process

## Abstract

This article presents the results of testing the conditions of closing foundry voids during the hot forging operation of an ingot made of zirconium with 1% Nb alloy and use of physical and numerical modeling, continuing research presented in a previous thematically related article published in the journal *Materials*. The study of the impact of forging operation parameters on the rheology of zirconium with 1% Nb alloy was carried out on a Gleeble 3800 device. Using the commercial FORGE^®^NxT 2.1 program, a numerical analysis was performed of the influence of thermo-mechanical parameters of the hot elongation operation in trapezoidal flat and rhombic trapezoidal anvils on the closure of foundry voids. The analysis of the obtained test results was used to formulate recommendations on the technology of hot forging and the anvilgeometry, ensuring closure of foundry voids. Based on their research, the authors conclude that the shape of the deformation basin and the value and hydrostatic pressure have the greatest influences on the closure of foundry voids.

## 1. Introduction

Zirconium alloy products, due to their physical and mechanical properties, are used in industries in which high requirements are placed on mechanical properties and corrosion resistance in aggressive environments. One of the main areas of use for zirconium alloy products is as construction elements for the production of fuel assemblies (nuclear reactor cores). The use of alloys on baize zirconia for the manufacture of the construction elements of poly rods is related to the small active cross-section for the absorption of thermal neutrons. They are characterized by the following additive properties: impermeability of fission products (compared to steel, they have a 15-fold smaller active cross-section for neutron capture) and resistance to radiation damage; low activity under neutron irradiation; sufficient mechanical strength;a low creep rate at high temperatures; sufficient thermal conductivity;andvery slow reactions with fuel and coolant (water), i.e., significant resistance to corrosion [1,2,3]. In addition to meeting the requirements for ensuring high mechanical and anti-corrosion properties, the products must have a uniform structure without metallurgical discontinuities [2]. The heterogeneity of the structure in the finished products may lead to stress concentrations, which may cause failure of fuel assemblies, which in turn may eventually lead to accidents [4].

There are many zirconium-based industrial alloys used in the nuclear industry. In general, the alloys are differentiated by the presence and content of elements such as Nb, Sn, Fe, Cr, Ni, and O. The choice of an alloy is related to the purpose of a given constructional element for a specific type of reactor. PWR zirconium with 1% Nb(M5) is used for making the plugs for cladding tubes of reactor fuel assemblies [5].

The technology for manufacturing tubes and rods from Zr–1%Nb alloy includes the following operations (Figure 1) [2,6]. During the hot forging process, the cast structure is fragmented, and the cross-section of the forging is reduced. The next stage is hot extrusion, during which there is a significant change in the cross-section. The next stage is cold rolling with intermediate and final annealing, during which a specific structure and the properties of the finished products are formed.

Authors [6,7] have stated that, in the process of ingot melting, the formation of a piping cavity in the axis of the ingot and of central porosity is inevitable. The formation of metallurgical discontinuities is related to the gradient of the crystallization rate in relation to the volume of the ingot. After melting, the ingots are subjected to ultrasonic testing to find metallurgical discontinuities. However, it is known from the authors’ experience that there may be central porosities with diameters of up to Ø10 mm in the axial part, as well as metallurgical voids with diameters of Ø1.0 mm at a certain distance from the axis of the ingot.

An analysis of the literature shows that methods for closing foundry voids during hot elongation of steel bars and aluminum and magnesium alloys are known [8,9,10,11,12,13,14,15], while no workswere found on closing foundry voids in the elongation operation of zirconium bars with the exception of the authors’ work [16].

Banaszek et al. [15] carried out numerical modeling of the closure of foundry voids during the forging operation of Mg alloys bars and physical verification of the obtained results.

In the work [15], the conditions were determined for closing foundry voids in the forging process of M5 alloy rods with the use of rhombic and flat anvils. To do so, the investigators used a physical simulator of metallurgical processes, the Gleeble 3800, and FORGE software, with which computer simulations of this process were carried out. The use of a combination of computer and physical modeling to determine the parameters of the hot elongation process is effective in terms of materials and time costs [14,17,18].

In the work, the authors showed the results of studies of the influence of the geometry of flat trapezoidal and rhombic trapezoidal anvils on the closure of foundry voids in the process of hot forging of M5 alloy rods. Based on the testing results, recommendations were made for the technology of hot forging processes, ensuring the closing of foundry voids occurring in ingots made of M5 alloy.

## 2. Purpose and Scope of Work

The objective of the study was to formulate recommendations for the technology of forging elongation that would ensure the closing of foundry voids occurring in zirconium alloy, observed after the vacuum process of double remelting in an electric arc furnace. The authors proposed that the operations of hot forging elongation should be performed with the use of flat trapezoidal and rhombic–trapezoidal tools. 

To achieve the goals of the study, tests investigatingthe forging of ingots in two anvil assemblies were carried out. Modeling of the elongation investigation was performed using the Forge program based on FEM, with the distributions of the hydrostatic pressure and effective strain being determined on a cross-section of the deformed zirconium alloy after each elongation process. 

To test the forging process, the datafrom plastometric tests of the zirconium alloy were obtained using the GLEBLLE 3800, based on which diagrams showing the relation between stress and real deformation were developed, and the coefficients of the plasticizing stress function were determined.

Based on the results of numerical studies, guidelines for the technology of forging shaped anvils for closing foundry voids were developed. 

The test results should help to improve the structural and mechanical properties of zirconium alloy during the appropriate selection of the tool shape and technological parameters of the forging process.

## 3. Methodology of Experimental Research

The zirconium alloy applied for the investigation was an M5 alloy with the following composition and chemical configuration: zirconium with 1.1% Nb, 0.05% Fe, and 0.6% O.

The impact of thermomechanical parameters of the hot elongation of the M5 alloy rod on the plasticizing stress *σ_p_* was determined using a Gleeble 3800 plastometer, 323 NY-355, Poestenkill, NY 12140, USA (Figure 2). Specimens of the M5 alloy in a crystallized state (Figure 3), each of a working portion with a diameter of 10 mm and a height of 12 mm, were used for the tests.

The tests were carried out in the temperature range of 770–950 °C [2,16]. On the basis of the Zr–Nb phase equilibrium scheme at the 870–950 °C temperature range, the structure of the metal is characterized by grains in the shape of the A2 unit cell. At forging temperatures of 750–870 °C, the structure of Zr is in the zone of α + β [2]. During the deformation of the zirconium with 1% Nb alloy sample in the specified temperatures, the limit value of the deformation decreases by half [19,20,21,22]. In this article, the authors show that the forging process of these alloys at temperatures less than 750 °C results in the occurrence of cracks.

Tests were carried out for the deformation speed range from 0.5 s^−1^ to 5.0 s^−1^, even though the average deformation speed for the hot elongation of the Zr alloys rods on the pressure is 0.5 s^−1^ [16]. However, in local places of the deformed rods in the shaped tools, the deformation speed is slightly higher, which is why it was decided to expand the research area.

The functionality of this process is the ability to obtain relatively large strain values (up to ε = 1.2 [19]); moreover, it is the most favorable strain state for describing the properties of the alloys during plastic deformation [17].

Origin software was used to process the final test results. This software allowed for a comprehensive analysis of the experimental data obtained, reducing the number of samples tested to two for a single point with an error limit less than 2–3%.

To use of the real results from plastometric tests of the M5 Zr alloys, an approximation of flow curves $\sigma_{P}-\varepsilon$ was carried out with the use of the generalizing connection–functions of Henzel A. and SpittelT. [19]:(1)$$\sigma_{P}=A\cdot e^{m1\cdot T}\cdot T^{m9}\cdot \varepsilon^{m2}\cdot e^{m4/\varepsilon }\cdot {\left(1+\varepsilon \right)}^{m5\cdot T}\cdot e^{m7\cdot \varepsilon }\cdot {\dot{\varepsilon }}^{m3}\cdot {\dot{\varepsilon }}^{m8\cdot T}$$
where $\sigma_{p}$ is stress, Tis temperature, ε is strain, $\dot{\varepsilon }$ is strain rate, A, and m1–m9 are function coefficients.

The approximation of the testing effects was carried out according to the method [14] in FORGE@2023 software, Transvalor S.A., E-Golf Park, 950 avenue Roumanille, CS 40237 Biot, 06904 Sophia Antipolis cedex, France.

## 4. Testing Results

Figure 4, Figure 5 and Figure 6 illustrate the $\sigma_{P}-\varepsilon$ curves of the zirconium alloys deformed at T = 770–950 °C for the deformation speed $\dot{\varepsilon }$ from 0.5 s^−1^ to 5.0 s^−1^. These figures show increases in the temperature of test T from 770 °C to 950 °C, during which the value of the plasticizing stress σ_p_ decreases about 2.5 times. An increase in the deformation speed from 0.5 s^−1^ to 5.0 s^−1^ led to an increase in the plasticizing stress σ_p_, and at T = 770 °C, this increase amounted to ~29%. At T = 850 °C, this increase was ~36%, and at T = 950 °C, it was 43%. The conducted research showed that the increase in the deformation speed affected the plasticizing stress σ_p_, which increased, and in addition, the magnitude of this increase was correlated with the temperature.

The $\sigma_{P}-\varepsilon$ plasticity curves for the tested temperature range varied. For the deformation speed $\dot{\varepsilon }$ = 0.5 s^−1^ in the tested temperature range, the $\sigma_{P}-\varepsilon$ curves reached the max. stress value for a programmed deformation value, and as the temperature increased, the max. plasticizing stress value σ_p_ moved toward actual smaller values.

The data shown in Figure 5 demonstrate that the curve obtained at deformation speeds of $\dot{\varepsilon }$ = 0.5 s^−1^ and $\dot{\varepsilon }$ = 5.0 s^−1^ are of a different character. The max. value of yield stress during deformation speed $\dot{\varepsilon }$ = 0.5 s^−1^ was observed for the value of real deformation ε = 0.12, and with a continued increase in deformation, a decrease in the value of yield stress was obtained. For the deformation speed $\dot{\varepsilon }$ = 5 s^−1^, the value of yield stress increases to the value of real deformation ε = 0.3, where the curve flattening effect takes place.

**Figure 5 materials-16-05431-f005:**
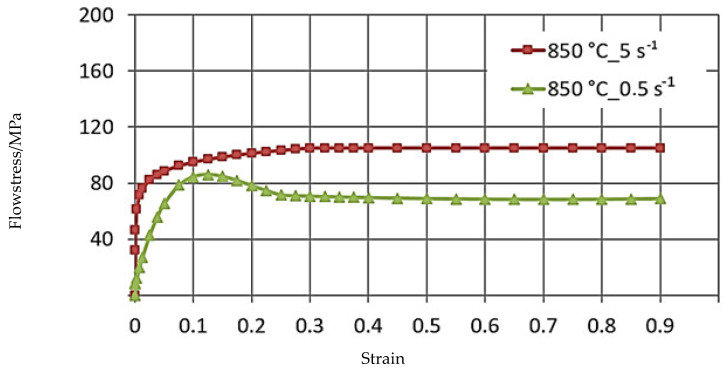
The $\sigma_{P}-\varepsilon$ flow curves of the M5 alloy, obtained for the temperature T = 850 °C in the deformation speed range $\dot{\varepsilon }$ from 0.5 to 5.0 s^−1^ using the Gleeble 3800 metallurgical simulator.

This difference may be due to the dynamic softening process being delayed with the increase in strain rate. Dynamic recrystallization is inhibited, and the softening of the material proceeds in accordance with the mechanism of dynamic polygonization. At the temperature T = 862 °C, a polymorphic transformation takes place in zircon, and the crystal lattice changes from the hexagonal A3 lattice to the spatially centered cubic lattice A2 [2]. The non-monotonicity of the curve may be caused by the beginning of the polymorphic transformation [11].

**Figure 6 materials-16-05431-f006:**
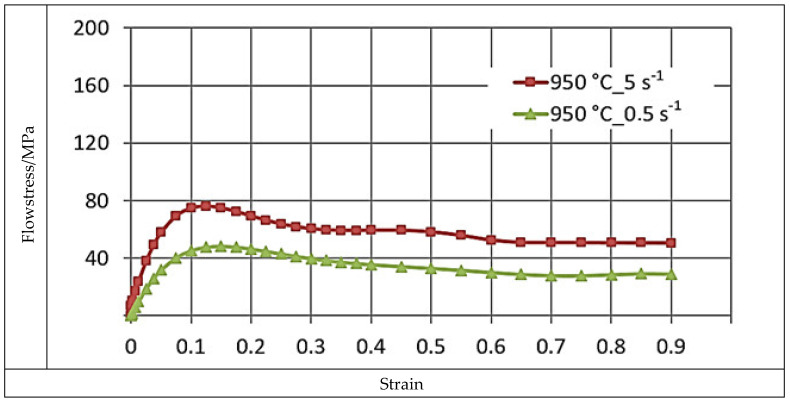
The $\sigma_{P}-\varepsilon$ flow curves of the M5 alloy, obtained for T = 950 °C in the deformation speed range $\dot{\varepsilon }$ at 0.5 to 5.0 s^−1^ using the Gleeble 3800.

As an effect of approximation of the investigational zirconium alloy curves, Hensel–Spittel coefficients were determined for the temperature and speed of the following plastic processing conditions: T = 770–950 °C, and $\dot{\varepsilon }$ = 0.5–5.0 s^−1^ (Figure 7).

The multifactorial Equation (1), using the coefficients shown in Figure 8 in the process of computer simulation, describes the changes in the value of the plasticizing stress of the M5 alloy depending on the thermomechanical parameters of the tested technological processes. The approximation results for the M5 alloy in the temperature range of T = 770–950 °C and deformation rates from 0.5 s^−1^ to 5.0 s^−1^ are shown in Figure 8 in the form of three-dimensional graphs.

It can be assumed that the coefficients of the approximating Function (1) are wellchosen if the average error does not exceed 15% [18]. For the range of parameters examined in the article, the average error of approximation is ~10%.

## 5. Methodology of Conducted Numerical Research

This article provides a theoretical analysis of the process of elongating Zr alloy bars for two different anvils to determine the possibility of closing foundry voids in Zr alloys. Tools were characterized by various geometrical surfaces of the deformation valley, which resulted in obtaining various directions and orientations of the pressure and friction forces in the deformed Zr alloy in the deformation place. The various directions and orientations of forces influenced the mechanism of welding of the foundry voids inside the deformed bars. Constant parameters of the forging operation were assumed, such as a forging initiation temperature of 950 °C, a value of relative compressionof 35%, and a value of the feed angleof 8 mm/s [23,24].

The parameters for the lengthening operation were adopted based on the authors’ experience in this field [15,16].

The computer program FORGE was applied to model the forging process in shaped anvils [25]. Forges enable the runningofthermomechanical simulations of, among other processes, plastic forming processes [26]. A description of the temperature, force, stress, strain, and thermomechanical and frictional laws used in the investigation appeared in [21].

For thermal calculations, the Galerkin equation was used, while the strengthening curves were approximated by the Henzel–Spitel equation (Figure 8).In this paper, to simulate the elongation operation, a thermo–visco–plastic model of the deformed body, which is based on the theory of large plastic deformations, was used. To generate the grid of finite elements, tetrahedral elements with the bases of triangles were used. In the generated model input, a number of nodes equal to 8958 was used for the simulation, while the number of tetrahedral elements adopted for the simulation was 40,760. The coefficient of friction between the surface of the tools and the bars, adoptedaccording to Coulomb’s law, was μ = 0.3. The value of the friction coefficient used for numerical modeling wasnot determined experimentally, but results from forging practice were used for many years.For the free forging of steels and alloys, friction coefficient values between 0.24 and 0.46 are assumed.

The following boundary conditions were assumed during the numerical tests: Indirect heat transfer between material and tool –“Steel Hot Medium.tef”; Friction coefficient between material and tools–“High.tff”;and Heat transfer coefficient between material and environment—“Air. tef”. The heat transfer coefficient connecting the tools and the zirconium alloy was assumed to be α = 10,000 W/m^2^K and the heat transfer coefficient between the zirconium alloy and the environment to be equal to λ = 10 W/m^2^K. The environment temperature was 25 °C, and the tool temperature was 250 °C. The starting temperature was 950 °C. In all forging steps, a relative 35% crumple was assumed. The speed of the upper tool was v = 8 mm/s, while the lower tool was assumed to be stationary.

Table 1 shows the values of the boundary and initial conditions used in the numerical model.

In the FORGE computer program, the diffusion model does not exist. The inference of foundry voids closing is based on the values of hydrostatic pressure and the temperature of the rods being elongated. Therefore, in the forging process, the aim is to achieve the maximum possible values of hydrostatic pressure within the foundry voids, as well as to maintain a high temperature close to that at the beginning of forging. Through the process of closing the foundry voids, there are high values of hydrostatic pressure on their right and left sides, while there is no hydrostatic pressure on their top and bottom sides (positive values of average stress) [13].

Zr alloy bars with a diameter of 100 mm and a length of l = 50 mm were deformed in two compositions of tools in four forging transitions.

The contour and size of the tools used in the investigation are shown in Figure 9 and Figure 10, respectively.

This work describes a graph of the forging operation for flat trapezoidal tools (Figure 9). During in first step, flat trapezoidal tools were used, in which a bar heated to a temperature of 950 °C was deformed with a relative reduction of 35%, and then the bar was rotated 90°. In the next step, with the bar using the same anvils, relative reduction of 35% was again applied, and it was rotated 90° clockwise. Before the third step, the bar was heated to a temperature of 950 °C because, after the second step, the temperature of the bar decreased to 750 °C, and according to the practice of forging Zr alloys, it is impossible to carry out further steps of the forging process. Subsequently, the next steps were already carried out in flat tools with a relative reduction of 35%. After the third pass, the bar was rotated 90° clockwise.

For the rhombic trapezoidal tools (Figure 10), the procedure was analogous to the forging scheme described above.

The geometry and distribution of the modeled foundry voids on the end face of the model Zr alloy bar are shown in Figure 11.

The initial model for the forging process was a cylinder with a diameter of 100 mm and a length of 50 mm, in which nine foundry voids were modeled with a length of charge (50 mm); the first axial one had a diameter of 10 mm, and the other eight, with a diameter of 1 mm, were distributed around the circumference of the model charge 25 mm from its axis (Figure 11). The charge modeled in this way corresponded to the shape of an ingot obtained after double remelting in a vacuum in an arc furnace. Artificially modeled holes simulated internal casting discontinuities, such as central porosity and casting voids. Descriptions of these defects can be found in the literature [23,24,25,26].

The model inputs and tools were drawn with AutoCad software. The foundry voids were drawn as cylinders inside the model input. Using the various tools in the program, the foundry voids were interpreted by the AC program as holes. The model diagram with artificially simulated foundry voids was exported to the Forge program as a file with the “stl” extension. The same process was performed for all of the tools’ compositions. In Forge, using the stl@meshing and volume@meshing tools, a 2D triangular grid was useful to the input and artificially modeled foundry voids, followed by a 3D tetrahedral grid. In total, for the correct simulation of the closure of foundry voids and their good formation, the built-in tools FoldsDetection andSelfContactwere used. Due to the use of the shown tools during the FEM simulation in the places of welded foundry voids, there were no calculation errors or mesh deterioration. During welding, the foundry voids of the nodal points of the tetrahedral elements in close contact connected; thus, the modeled foundry voids were closed. During modeling, there were no errors in the form of interpenetration of nodes of tetrahedral elements around closed foundry voids.

On the basis of the simulations of the closing process of foundry voids during the forging of bars made of zirconium alloy in flat and rhombic tools, the volume values of the remaining unwelded foundry voids were determined. The procedure was as follows.After the simulation of the forging process was completed, the stlfile with a deformed bar was exported from the Forge program to the RinoCeros program, where the tetrahedral element grid was separated in this program, and the bar contour elements were removed. After this step, only tetrahedral elements that constituted the contours of the unwelded discontinuities remained. The next step was to merge the previously broken grid of tetrahedral elements. In the last step, using the physical parameters tool in the RinoCeros program, the whole volume of foundry voids for individual forging process was obtained.

After the bar forging process was carried out in two anvil compositions, the values of the hydrostatic pressure at the nodal points of the tetrahedra, constituting the contour of the unwelded foundry voids, were determined as the arithmetic means of the occurring stresses.The same procedure was performed for the value of the effective strain. All the results shown in the article are subject to some errors resulting from the need to analyze the numerical and data and the results obtained.

## 6. Analysis of Distribution of Hydrostatic Pressure Values During Forging Process

The test effects concerning the hydrostatic pressure distribution during the zirconium alloy bar extension operation in two tool compositions are shown in Figure 12, Figure 13, Figure 14, Figure 15, Figure 16, Figure 17, Figure 18, Figure 19, Figure 20, Figure 21, Figure 22, Figure 23, Figure 24, Figure 25, Figure 26 and Figure 27.

### 6.1. Analysis of the Distribution of Hydrostatic Pressure Values During the Forging Process in Flat Trapezoidal Anvils

Figure 12, Figure 13, Figure 14, Figure 15, Figure 16, Figure 17, Figure 18 and Figure 19 present the distributions of hydrostatic pressure values obtained during the numerical computation of elongating Zr alloy bars in flat trapezoidal anvils.

**Figure 12 materials-16-05431-f012:**
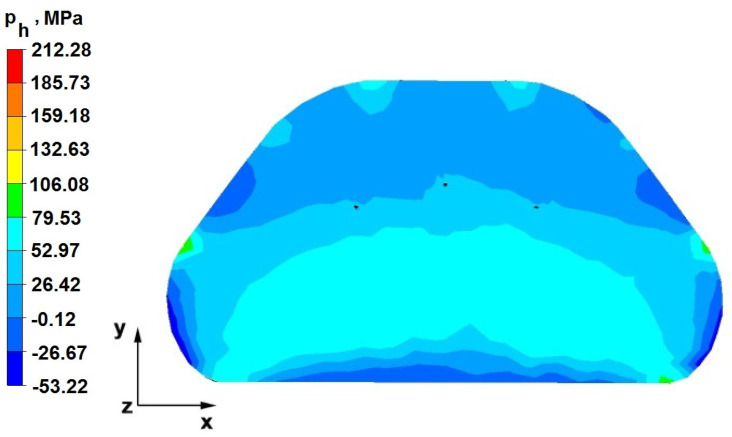
Hydrostatic pressure on the cross-section of a forged bar in the first step with a 35% crumple.

**Figure 13 materials-16-05431-f013:**
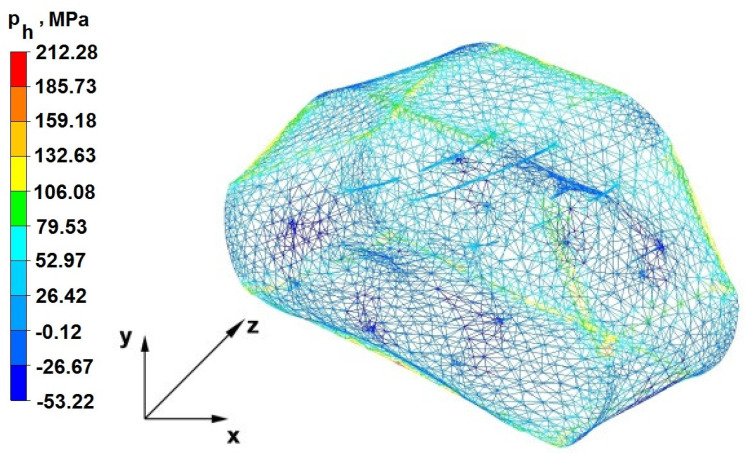
Hydrostatic pressuredistribution with a view of unclosed defects in the volume of a forged bar in the first step with a 35% crumple.

**Figure 14 materials-16-05431-f014:**
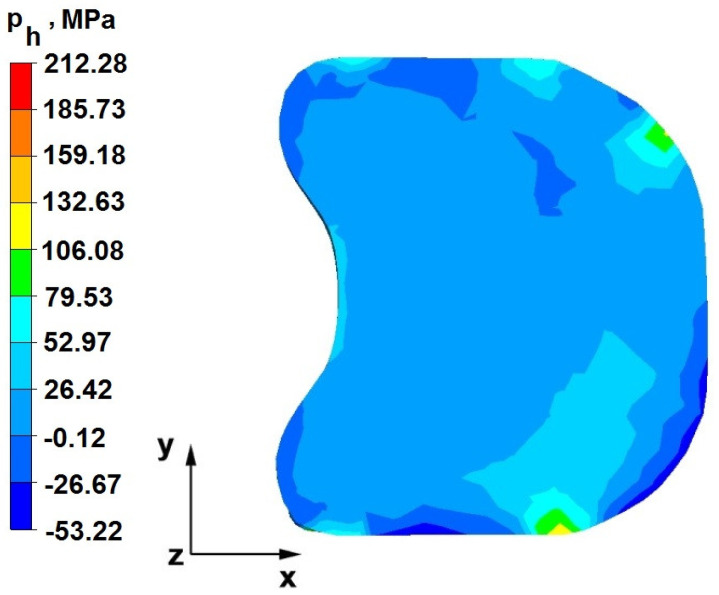
Hydrostatic pressure on the cross-section of a forged bar in the second step after turning through an angle of 90° with a 35% crumple.

**Figure 15 materials-16-05431-f015:**
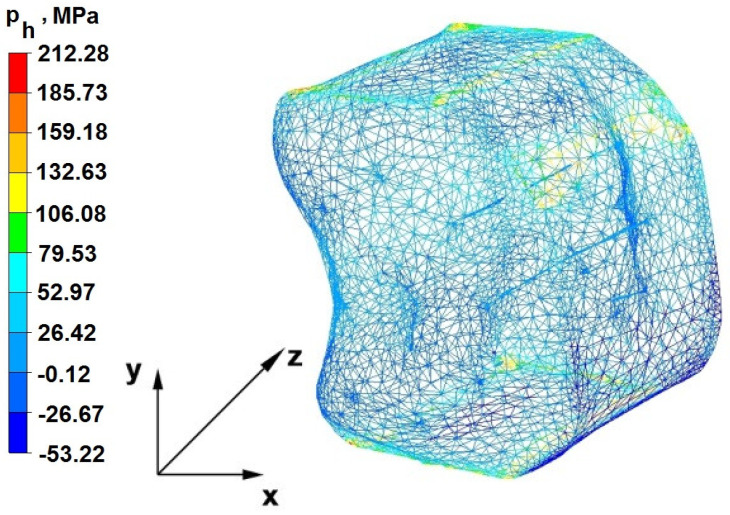
Hydrostatic pressuredistribution with a view of unclosed defects in the volume of a forged rod in the second transition after a rotation of 90° with 35% crumple.

**Figure 16 materials-16-05431-f016:**
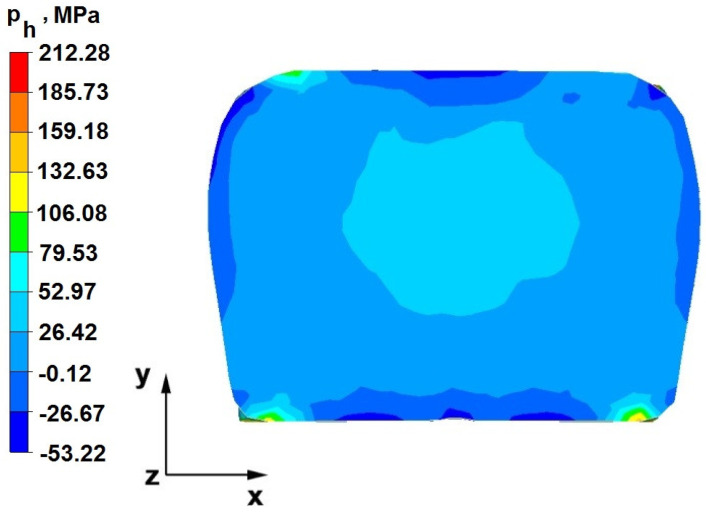
Hydrostatic pressure on the cross-section of a forged bar in the third step after turning through an angle of 90° with a 35% crumple (change to flat anvils).

**Figure 17 materials-16-05431-f017:**
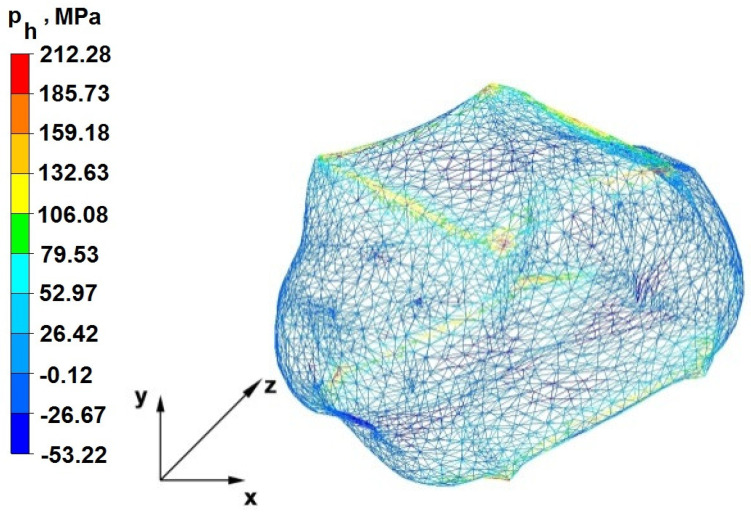
Hydrostatic pressuredistribution with a view of unclosed defects in the volume of a forged rod in the third transition after a rotation of 90° with 35% crumple(change to flat anvils).

**Figure 18 materials-16-05431-f018:**
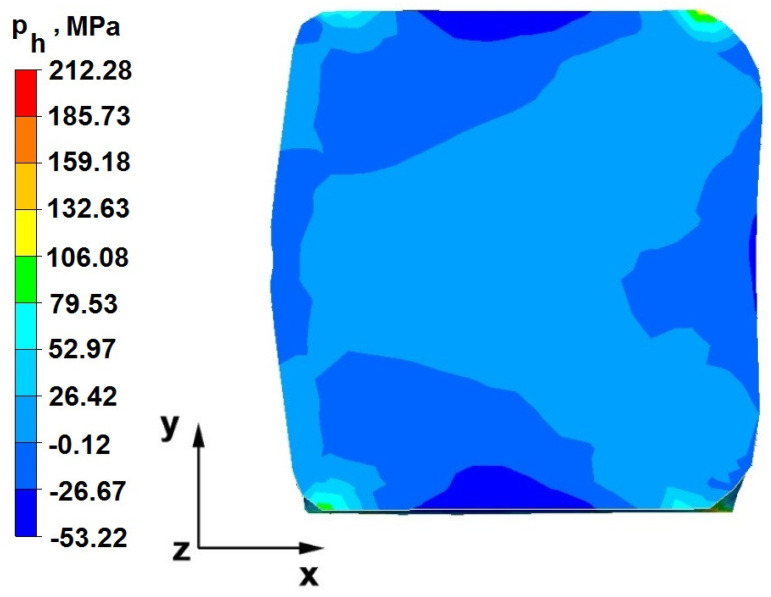
Hydrostatic pressure on the cross-section of a forged bar in the fourth step after turning through an angle of 90° with a 35% crumple (change to flat anvils).

**Figure 19 materials-16-05431-f019:**
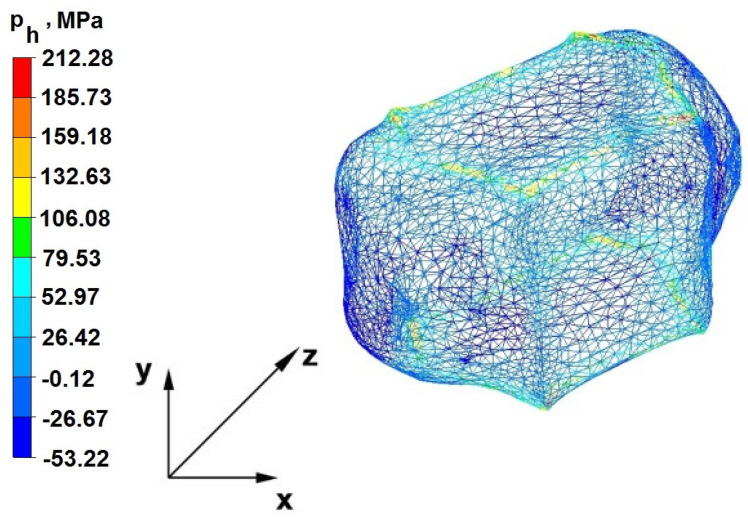
Hydrostatic pressuredistribution with a view of unclosed defects in the volume of a forged rod in thorough transition after a rotation of 90° with 35% crumple (change to flat anvils).

The data in Figure 12 and Figure 13 show that the axial foundry voids with a diameter of 10 mm were welded in the first forging step, indicating that the use of flat trapezoidal tools results in obtaining the appropriate characteristics of the stress distribution in the axial zones of the deformed rod, with a positive effect on the closing discontinuities. The value of hydrostatic pressure in the axis of the forged bar was 53 MPa. This outcome is especially important because, in the first steps of the forging process, the temperature of the deformed bar does not change, which facilitates welding of the discontinuities. It is also worth noting that, after the casting process for ingot molds, the discontinuities in the axis of the ingot have the largest volume, and they are difficult to weld in the further stages of the elongation process. Therefore, it is very important that the compressive stresses of the highest possible values occur in these zones. The use of a flat trapezoidal anvil in the starting bar forging process also favors the process of welding foundry voids in the areas under the operation of the lower flat tool because, as shown by the hydrostatic pressure distribution (Figure 12), its values are within the range of 26–53 MPa. On the other hand, the forging process in flat trapezoidal anvils does not favor the welding of foundry voids located in places under the action of the upper trapezoidal form. The values of compressive stresses in this place range from 5 to 25 MPa. There were three unwelded foundry voids left.

Based on the test results shown in Figure 14 and Figure 15, it can be concluded that, in the predominant volume of the deformed rod, adecreasein the value of the hydrostatic pressure from 53 MPa to a value close to zero was noted, proving its absence. The lack of hydrostatic pressure does not favor the welding of the foundry voids. The only area where the hydrostatic pressure was still present (its value was 26 MPa) was the area on the right, impacted by the lower anvil. Two consecutive discontinuities did not fully weld in the second forging step due to a lack of hydrostatic pressure in the place of their occurrence, and the value of the average stress was 0.12 MPa there; i.e., it was tensile stress, unfavorable for the welding of the discontinuities.

Figure 16 and Figure 17 show the test results obtained after deformation in the third forging step, where the flat trapezoidal tools were replaced with flat tools, and the deformed bar was heated to the starting temperature. The data analysis shows that no unwelded foundry voids were observed in the volume of the deformed bar. In the central deformation zone of the rod, the hydrostatic pressure was 26 MPa. Outside of this zone, there was no hydrostatic pressure.

The data in Figure 18 and Figure 19 show that, in the fourth and last elongation step in the entire volume of the deformed rod, no hydrostatic pressure was recorded outside the corner areas. There were positive mean stress values in the range of 0.12–53 MPa. The introduction of flat anvils on the last stages of rod formation, after using the trapezoidal flat anvil to introduce high hydrostatic pressure, did not bring the intended effect. During the forging with flat anvils, positive values of average stresses in most areas of the forged rod were observed, preventing the welding of discontinuities, especially those with large initial dimensions.

### 6.2. Analysis of the Distribution of Hydrostatic Pressure Values in the Forging Process in the Rhombic Trapezoidal Tools

Figure 20, Figure 21, Figure 22, Figure 23, Figure 24, Figure 25, Figure 26 and Figure 27 show the distribution of hydrostatic pressure values obtained in the numerical computation of the elongation of the bar from the zirconium alloy in rhombic trapezoidal tools.

**Figure 20 materials-16-05431-f020:**
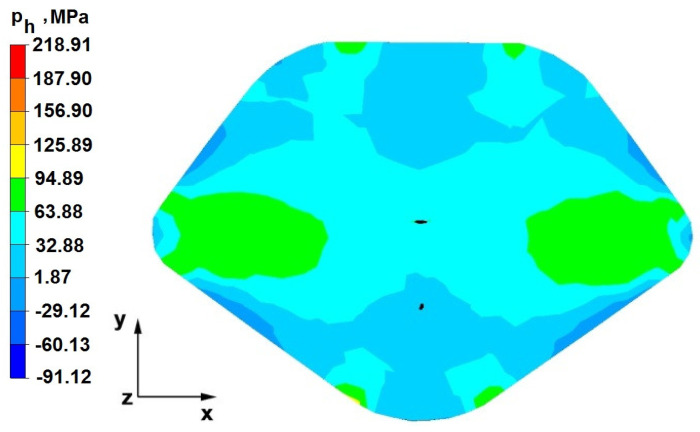
Hydrostatic pressure on the cross-section of a forged bar in the first step with a 35% crumple.

**Figure 21 materials-16-05431-f021:**
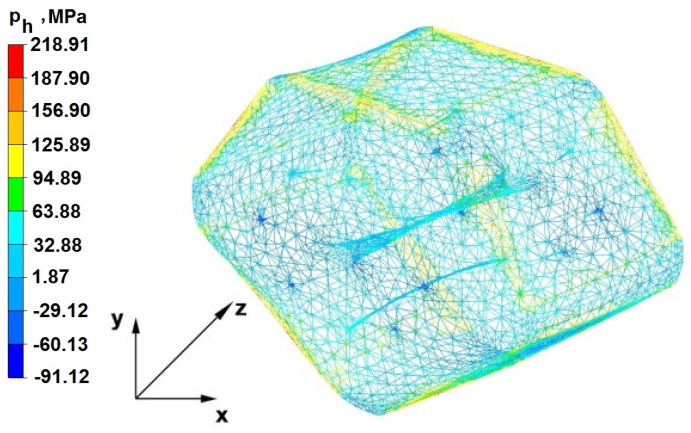
Hydrostatic pressure distribution, along with the view of unclosed defects in the volume of a forged bar in the first step with a 35% crumple.

**Figure 22 materials-16-05431-f022:**
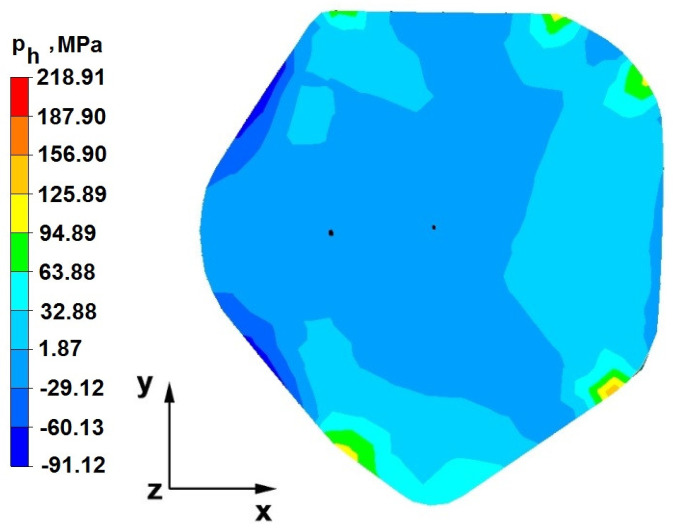
Hydrostatic pressure on the cross-section of a forged bar in the second pass after turning through an angle of 90° with a 35% crumple.

**Figure 23 materials-16-05431-f023:**
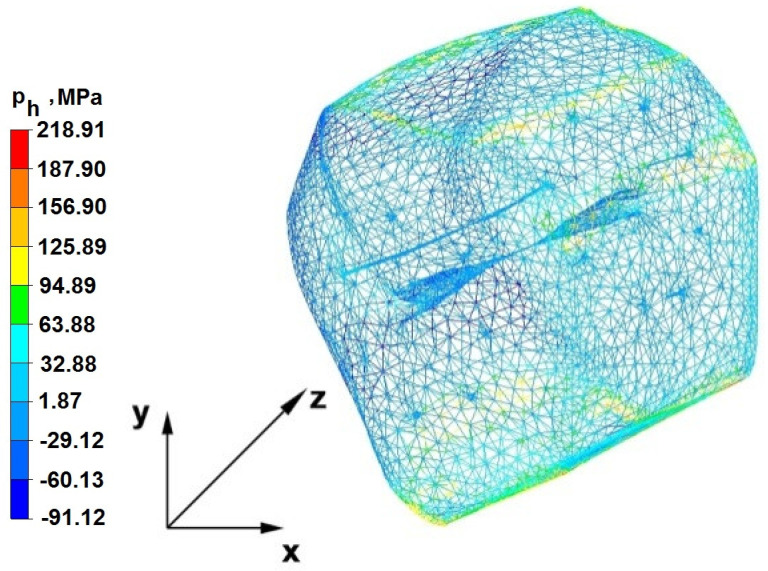
Hydrostatic pressure distribution along with the view of unclosed defects in the volume of a forged rod in the second pass after turning through an angle of 90° with a 35% crumple.

**Figure 24 materials-16-05431-f024:**
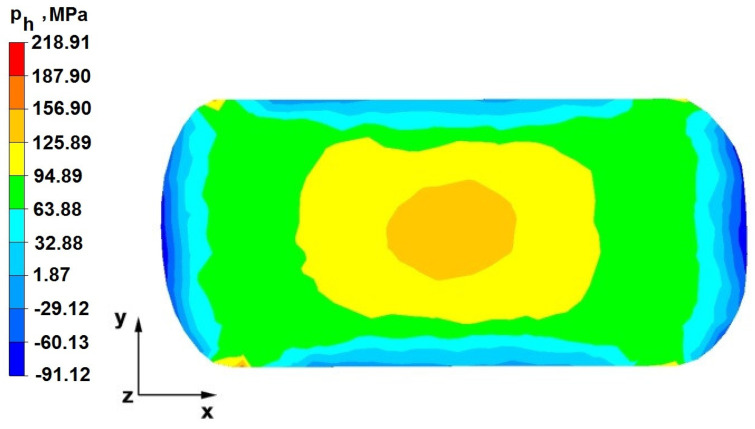
Hydrostatic pressure on the cross-section of a forged bar in the third pass after turning through an angle of 90° with a 35% crumple(change to flat anvils).

**Figure 25 materials-16-05431-f025:**
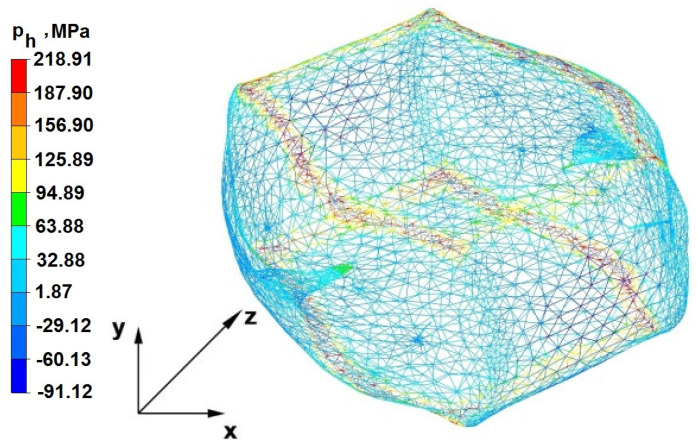
Hydrostatic pressure distribution, along with the view of unclosed defects in the volume of a forged rod in the third transition after turning through an angle of 90° with a 35% crumple(change to flat anvils).

**Figure 26 materials-16-05431-f026:**
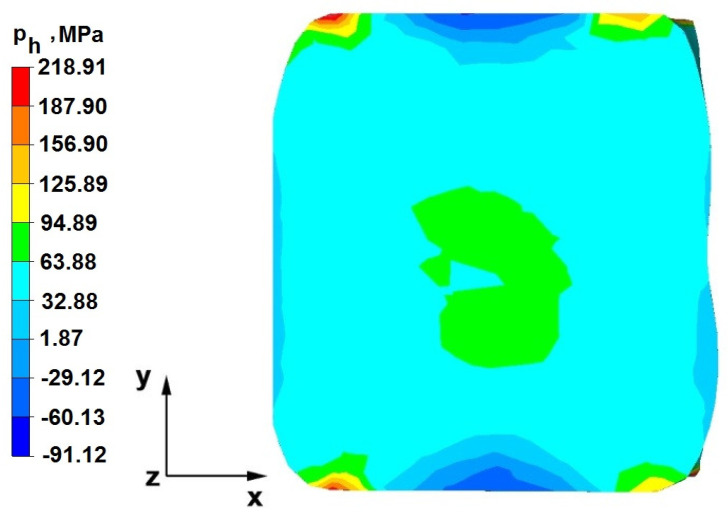
Hydrostatic pressure on the cross-section of a forged bar in the fourth pass after turning through an angle of 90° with a 35% crumple(change to flat anvils).

**Figure 27 materials-16-05431-f027:**
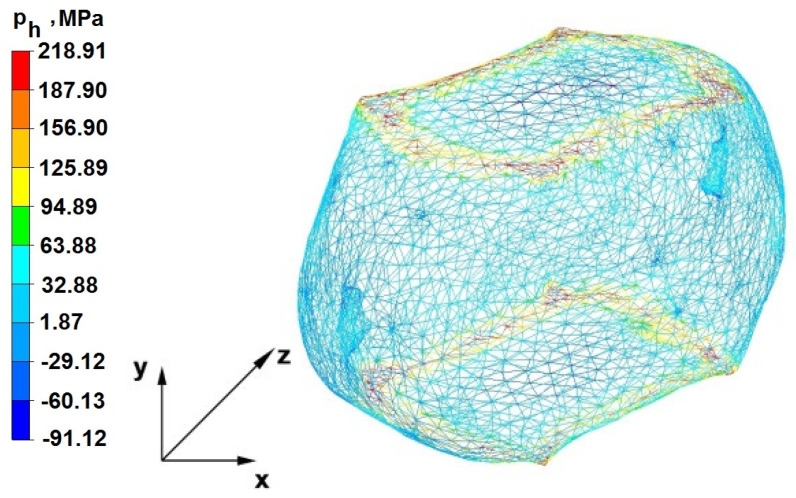
Hydrostatic pressure distribution, along with a view of unclosed defects in the volume of a forged bar in the fourth transition after turning through an angle of 90° with a 35% crumple(change to flat anvils).

The data in Figure 20 and Figure 21 show that, during the implementation of the initial stages of the elongation using rhombic trapezoidal anvils, the axial discontinuity, as well as the discontinuity lying in the zone of impact of the lower rhombic anvil, is not completely welded. In the place of axial discontinuity occurrence, the hydrostatic pressure was 33 MPa, while in the place of the second occurrence ofunclosed foundry voids, it was 2 MPa. The pressure value obtained in this place was too low for the process of closing the foundry voids. The value of the hydrostatic pressure of 64 MPa is sufficient to weld the discontinuities in the corners of the deformed rod along the x axis.

By analyzing the data presented in Figure 22 and Figure 23, it can be concluded that, in the majority of the volume of the deformed bar in the next forging step, no hydrostatic pressure occurred; therefore, the foundry voids remaining after the first forging step were also not closed in the second step. In these places, no hydrostatic pressure existed, and the values of positive mean stress ranged from 29 to 60 MPa. This outcome proves that, in this area, tensile stresses occurred, blocking the closing process of foundry voids.

Figure 24 and Figure 25 show the distribution of hydrostatic pressure on the cross-section of a deformed zirconium alloy rod previously heated to the initial forging temperature after the third step, made with flat tools with a 35% crumple. No unclosing foundry voids were observed because, in almost the entire volume of the deformed bar, the hydrostatic pressure achieved values of 64–126 MPa, creating favorable conditions for the closing of foundry voids.

Based on the data in Figure 26 and Figure 27, it can be seen that, after the fourth, final step in the entire volume of the deformed bar, the value of the hydrostatic pressure was sufficiently large, influencing the closing of the foundry voids. The values of hydrostatic pressure occurring in the entire volume of the deformed bar ranged from 33 to 64 MPa.

## 7. The Distribution of Deformation Intensity During The Forging Process

To obtain the deliberate characteristics of the deformations that would facilitate closing of the foundry voids, the authors proposed that the forging process should be carried out in rhombic trapezoidal and flat trapezoidal anvils, allowing them to control the direction, orientation, and magnitude of the vectors of friction and the pressure forces acting in the deformation area and to select the appropriate values of the main technological parameters of the forging process.

The effects of the investigation of the distribution of hydrostatic pressure in the elongation of the zirconium alloy bar in flat, trapezoidal, andrhombic-trapezoidal anvils are presented in Figure 28, Figure 29, Figure 30, Figure 31, Figure 32, Figure 33, Figure 34, Figure 35, Figure 36, Figure 37, Figure 38, Figure 39, Figure 40, Figure 41, Figure 42 and Figure 43.

### 7.1. The Distribution of the Deformation Intensity Values in the Forging Process in Flat Trapezoidal Anvils

Figure 28, Figure 29, Figure 30, Figure 31, Figure 32, Figure 33, Figure 34 and Figure 35 show the effective strain distributions obtained during numerical computation of zirconium alloy bar elongation in flat trapezoidal anvils.

**Figure 28 materials-16-05431-f028:**
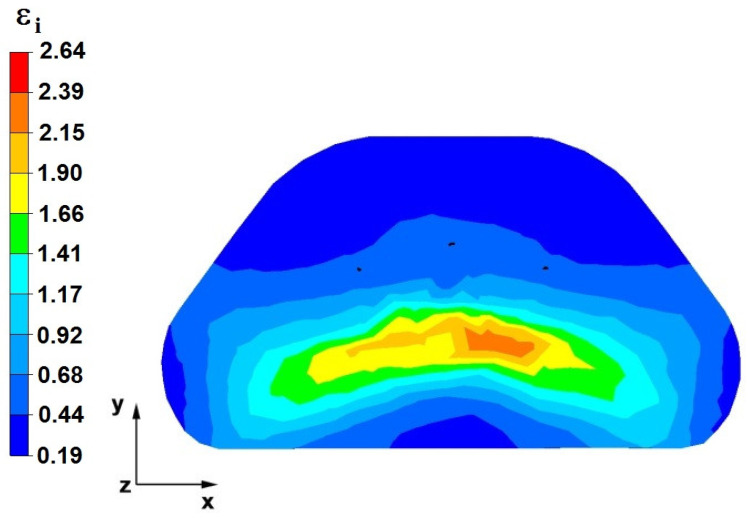
The effective strain on the cross-section of the forged bar in the first step with a 35% crumple.

**Figure 29 materials-16-05431-f029:**
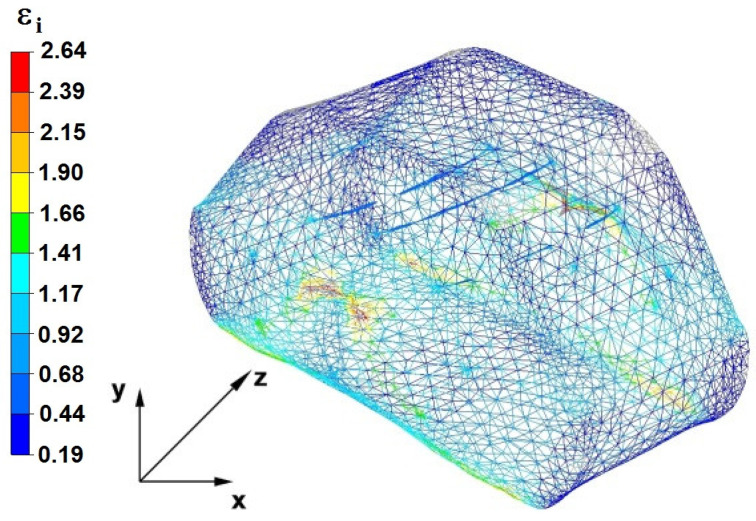
Distribution of the effective strain, along with a view of the unclosed defects in the volume of the forged bar in the first step with a 35% crumple.

**Figure 30 materials-16-05431-f030:**
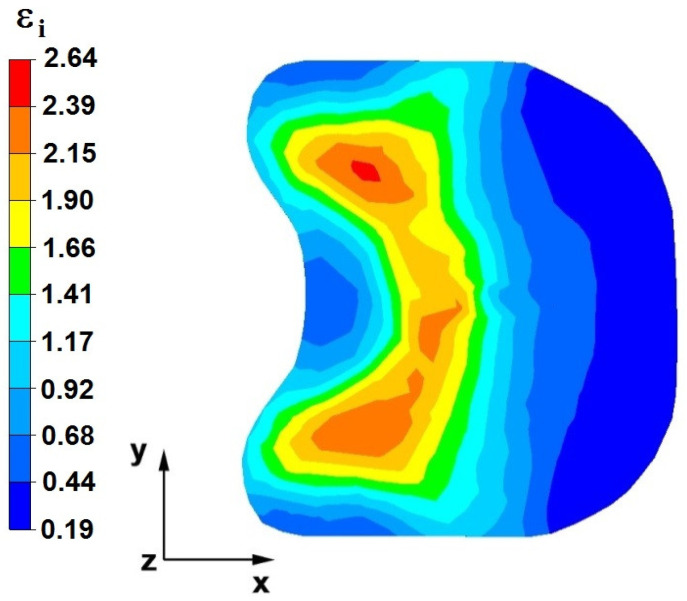
The effective strain on the cross-section of the forged bar in the second step after turning through an angle of 90° with a 35% crumple.

**Figure 31 materials-16-05431-f031:**
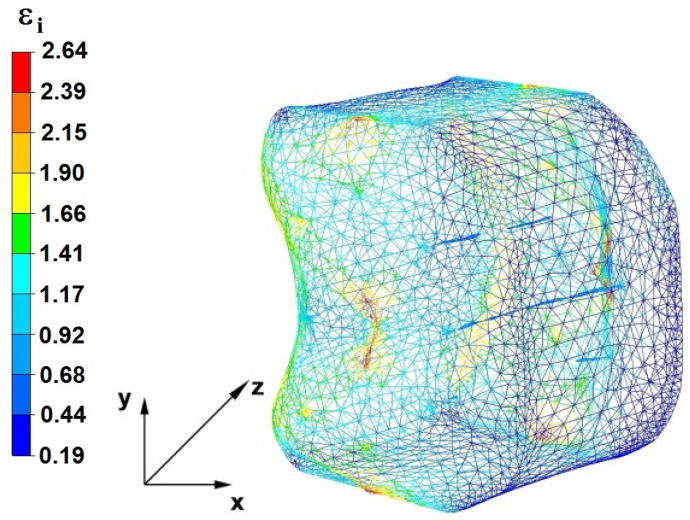
Distribution of the effective strain, along with a view of the unclosed defects in the volume of the forged bar in the second step after turning through an angle of 90° with a 35% crumple.

**Figure 32 materials-16-05431-f032:**
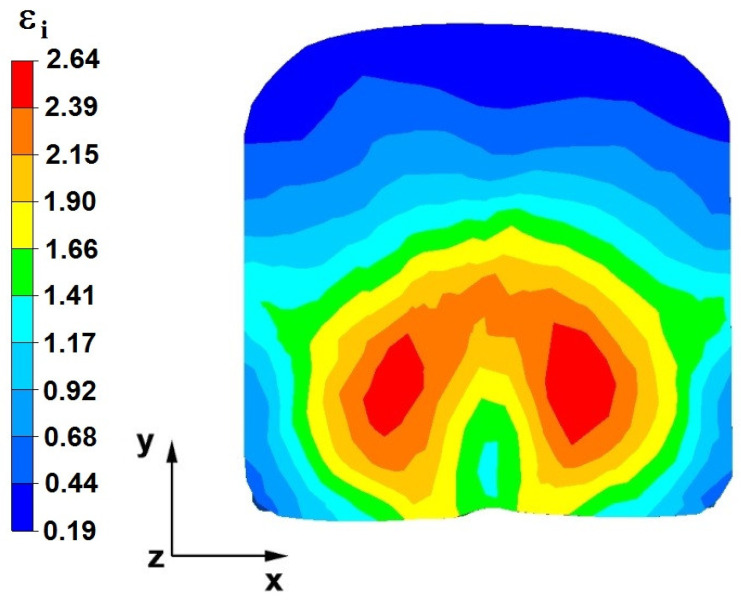
The effective strain on the cross-section of the forged bar in the third step after turning through an angle of 90° with a 35% crumple(change to flat anvils).

**Figure 33 materials-16-05431-f033:**
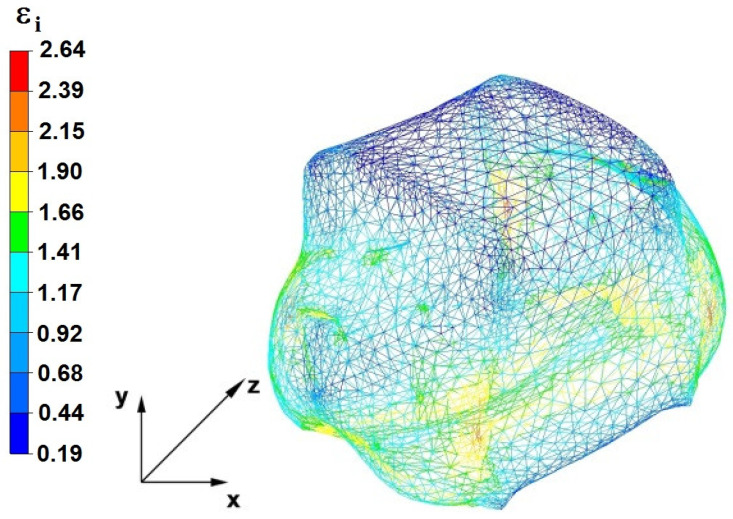
Distribution of the effective strain, along with a view of the unclosed defects in the volume of the forged bar in the third step after turning through an angle of 90° with a 35% crumple(change to flat anvils).

**Figure 34 materials-16-05431-f034:**
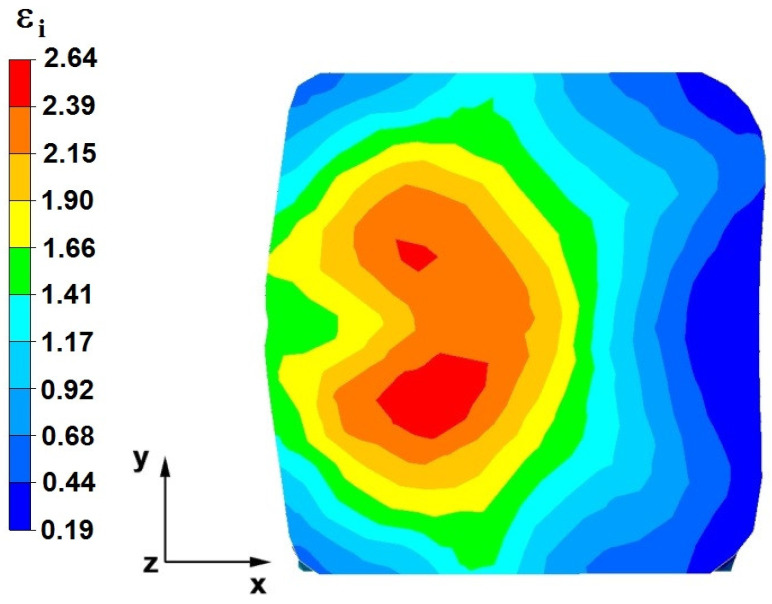
The effective strain on the cross-section of the forged bar in the fourth step after turning through an angle of90° with a 35% crumple(change to flat anvils).

**Figure 35 materials-16-05431-f035:**
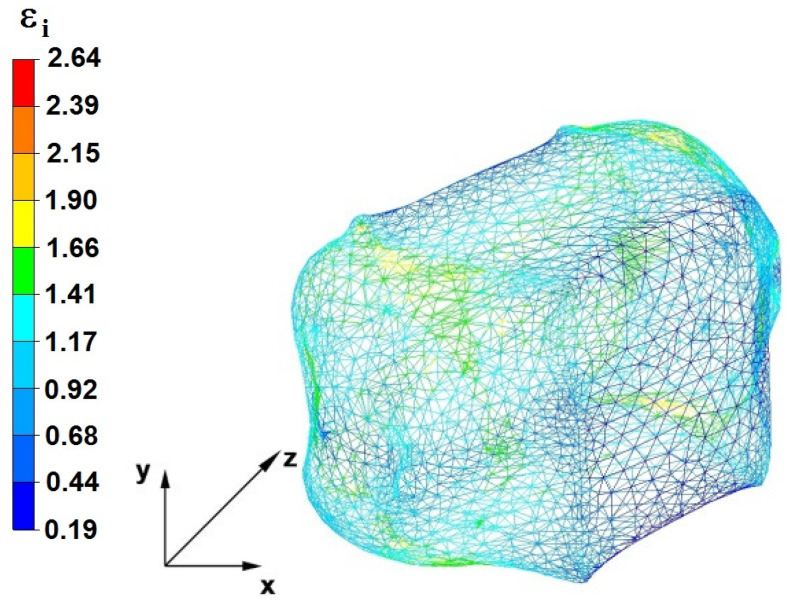
Distribution of the effective strain, along with a view of the unclosed defects in the volume of the forged bar in the fourth step after turning through an angle of 90° with a 35% crumple(change to flat anvils).

By analyzing the data in Figure 28 and Figure 29, it can be seen that the axial discontinuity, as well as the foundry voids under the direct influence of the lower flat anvil, was completely closed after the first forging step. A favorable state of deformation developed in the metal located in the deformation basin due to the appropriate directions and orientations of frictional forces and the pressure forces resulting from the impact of the employed surfaces of the shaped tools on the deformed rod. Such a distribution of forces caused the occurrence of large deformations in the middle and lower places of the deformed bar, leading to the effective strain values ranging from 1.17 to 2.64. Only the discontinuities in the areas of the metal impacted by the upper trapezoidal anvil remained completely unwelded. The values of the effective strain were too small there to allow for complete welding of the discontinuities, as they amounted to 0.19–0.44.

The data in Figure 30 and Figure 31 show that, after the second step, the foundry voids remaining after the first step were still not completely closed because, in the places of their occurrence, the values of the effective strain were too small and amounted to 0.19–0.44. On the other hand, in the area on the left side of the deformed bar, the effective strain values were favorable for the process of closing the foundry voids, as they were in the range of 1.41–2.64.

Figure 32 and Figure 33 show the effective strain values after the third step and after changing the shaped tools to flat tools, before heating the forged bar to the starting temperature and its rotation by an angle of 90°. The data presented in Figure 34 and Figure 35 show that the foundry voids remaining after the second forging step were completely closed. The values of the effective strain in the middle and lower places of the deformed bar were high and ranged from 1.41 to 2.64. On the other hand, the distribution of the effective strain values, which were in the range of 0.19–0.92 and thus unfavorable for the welding of discontinuities, was observed in the zone of the bar impacted by the upper anvil.

By analyzing the data in Figure 34 and Figure 35, it can be concluded that the values of the effective strain in the last (fourth) step slightly increased. Apart from the small zone to the right of the y axis of the deformed bar, these values were large and ranged between 1.17 and 2.64.

### 7.2. The Distribution of the Deformation Intensity Values in the Forging Process in the Rhombic Trapezoidal Tools

Figure 36, Figure 37, Figure 38, Figure 39, Figure 40, Figure 41, Figure 42 and Figure 43 show the effective strain distributions obtained in numerical computations of the zirconium alloy bar with rhombic trapezoidal tools.

**Figure 36 materials-16-05431-f036:**
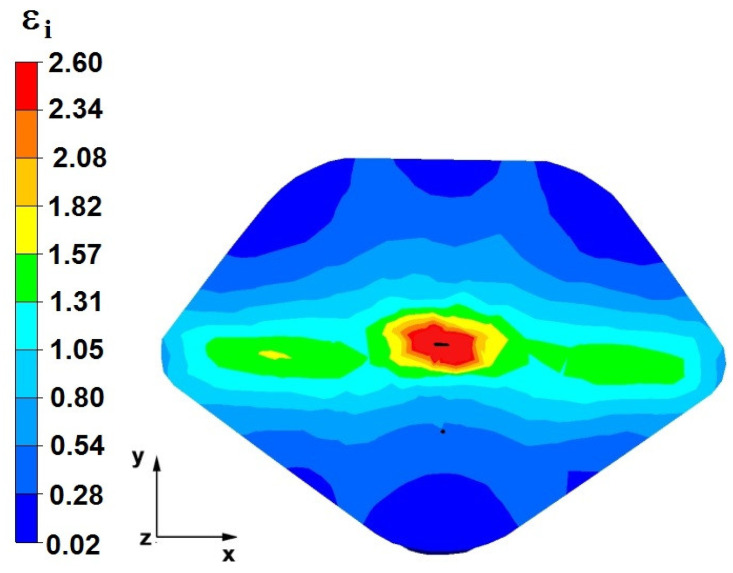
The effective strain on the cross-section of a forged bar in the first step with a 35% crumple.

**Figure 37 materials-16-05431-f037:**
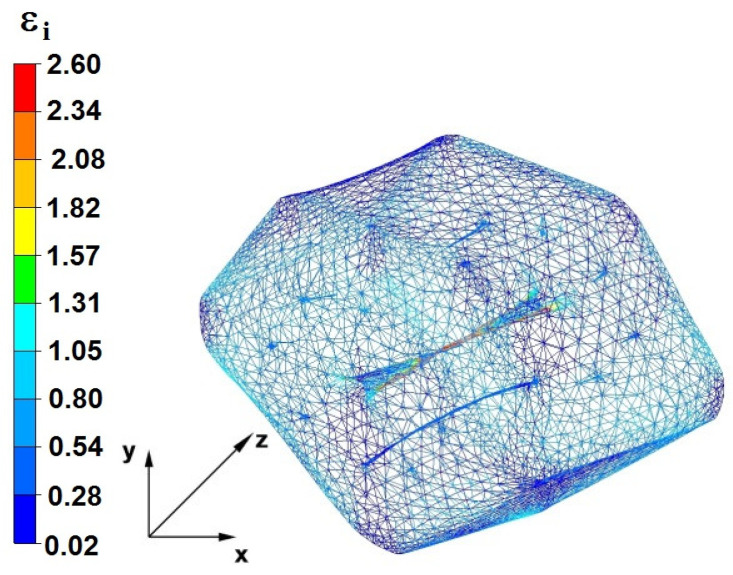
Deformation effective strain, along with the view of non-closed defects in the volume of the forged bar in the first step with a 35% crumple.

**Figure 38 materials-16-05431-f038:**
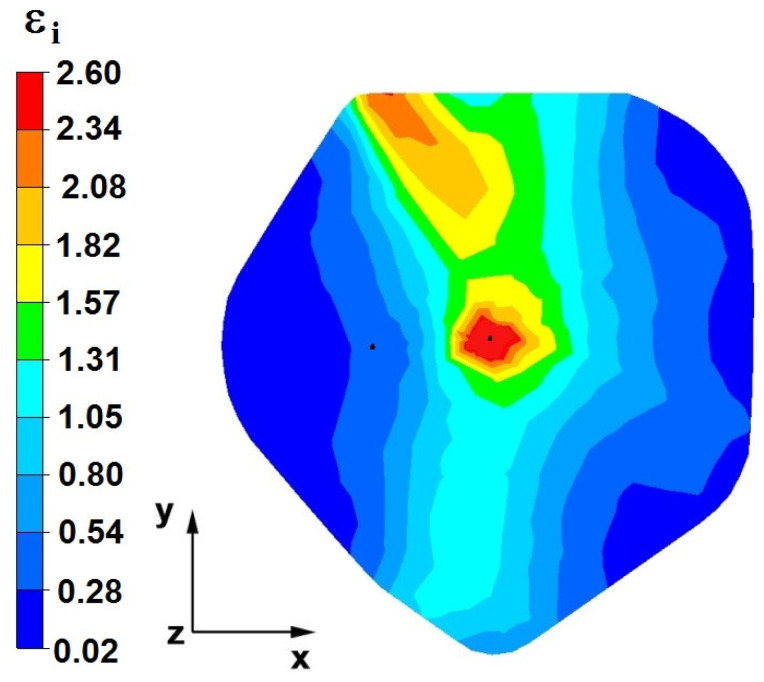
The effective strain on the cross-section of a forged bar in the second step after turning through an angle of 90° with a 35% crumple.

**Figure 39 materials-16-05431-f039:**
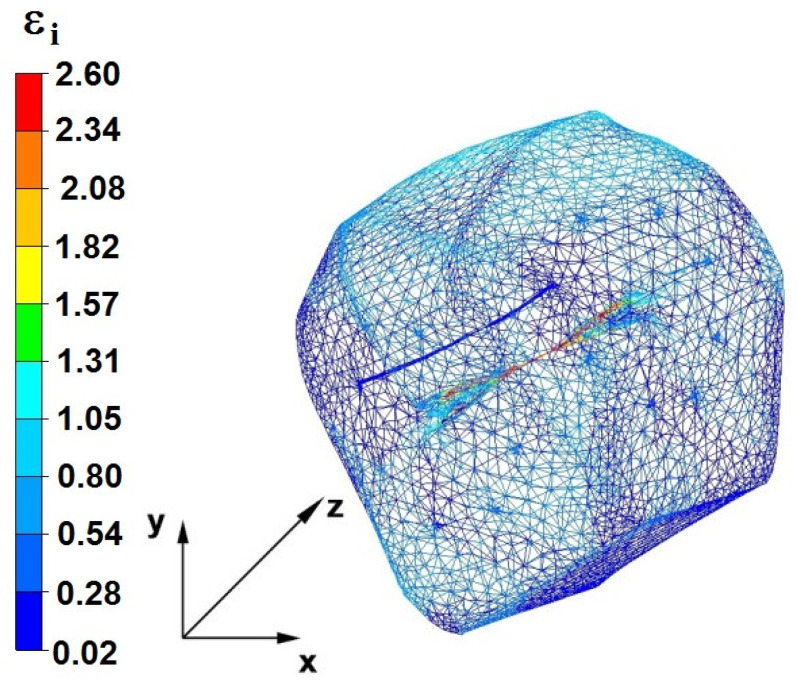
Deformation effective strain, along with the view of non-closed defects in the volume of the forged bar in the second step after turning through an angle of 90° with a 35% crumple.

**Figure 40 materials-16-05431-f040:**
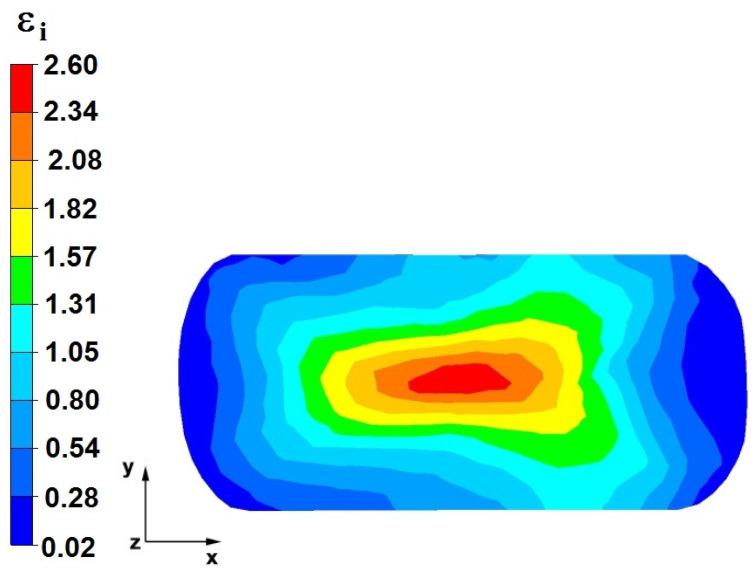
The effective strain on the cross-section of a forged bar in the third step after turning through an angle of 90° with a 35% crumple (change to flat anvils).

**Figure 41 materials-16-05431-f041:**
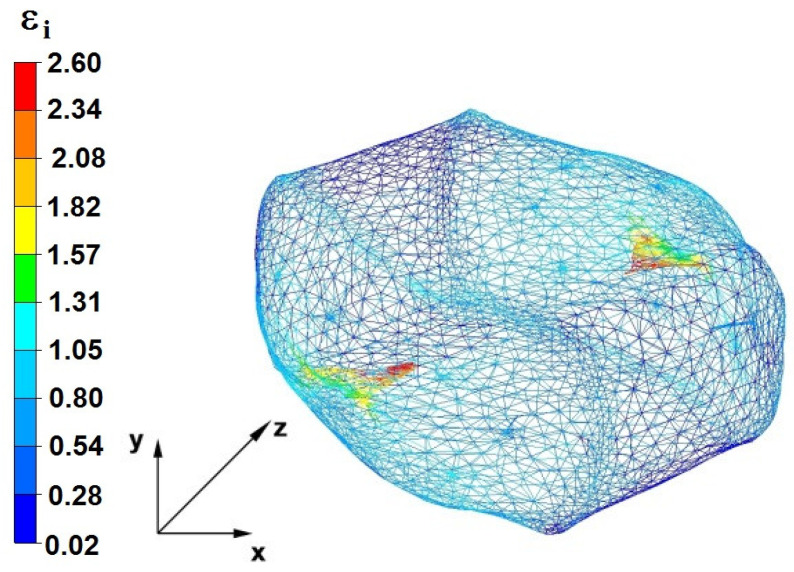
Deformation effective strain, along with the view of non-closed defects in the volume of the forged bar in the third step after turning through an angle of 90° with a 35% crumple (change to flat anvils).

**Figure 42 materials-16-05431-f042:**
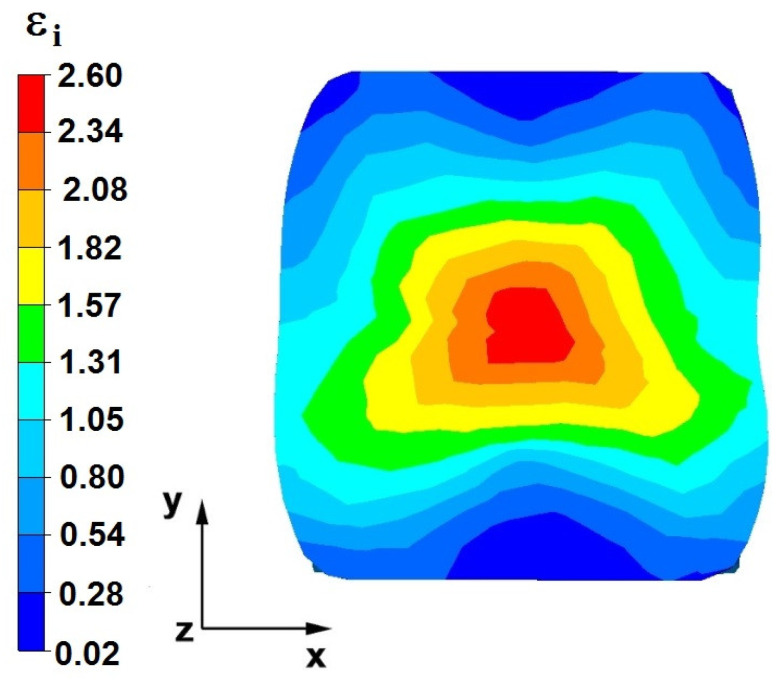
The effective strain on the cross-section of a forged bar in the fourth step after turning through an angle of 90° with a 35% crumple (change to flat anvils).

**Figure 43 materials-16-05431-f043:**
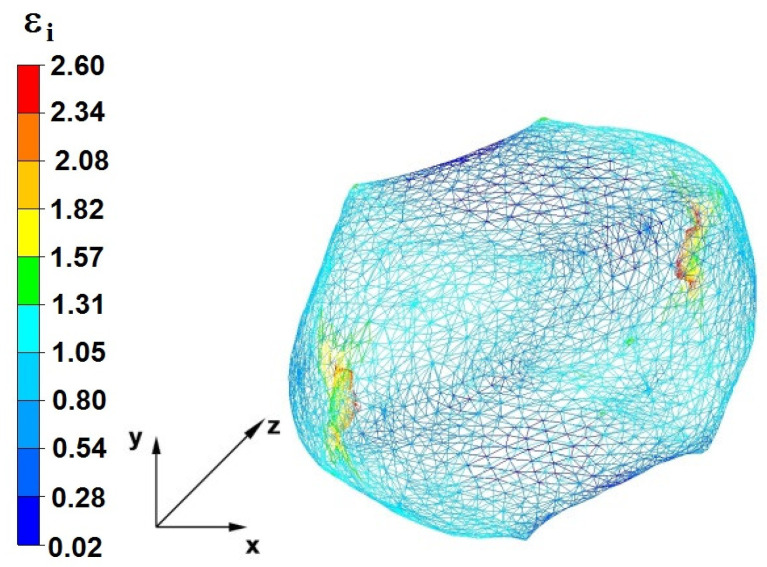
Deformation effective strain, along with the view of non-closed defects in the volume of the forged bar in the fourth step after turning through an angle of 90° with a 35% crumple (change to flat anvils).

By analyzing the data presented in Figure 36 and Figure 37, it can be seen that, in the central areas of the deformed bar, the value of the deformation intensity was 2.60, which was insufficient for the complete closing of the axial foundry voids present there. In the lower part of the bar, which was deformed after the first step, unclosed foundry voids were also observed because values of the effective strain in that zone were small, ranging from 0.28 to 0.54. The low intensity of deformations concentrated in the region of foundry voids did not have a positive effect on the closing process of foundry voids.

The data in Figure 38 and Figure 39 show that, due to the high value of the effective strain in the second forging transition, whichwas equal to 2.60, the axial foundry voids were partially welded. Additionally, the foundry voids, now located in the left part of the deformed bar because the bar had been turned 90°, was not closed due to the presence of a small value of the effective strain equal to 0.28.

From the data presented in Figure 40 and Figure 41, which present the distribution of the effective strain values after the third step, no modeled foundry voids were observed. In most of the cross-sectional zone of the deformed bar, there was a favorable distribution of the effective strain values within the range of 1.05–2.60. Additionally, based on the data presented in Figure 42 and Figure 43, it can be concluded that, on the cross-sectional area of the bar deformed in the fourth step, the values of the effective strain were large (in the range of 1.05–2.60) and facilitated the closure of foundry voids. The exception was the area at the contact surface of the alloy with the anvil, in which the effective strain values were small and fell within the range of 0.02–0.28. However, internal foundry voids never occur in the near surface zones.

## 8. The Influence of the Anvil Shape on Changes in the Volume of Discontinuities, Hydrostatic Pressure, and the Effective Strain in the Zr Alloy in the First Two Steps

Figure 44, Figure 45 and Figure 46 show diagrams of the total volume of unclosed foundry voids, mean values of hydrostatic pressure, and arithmetic mean values of the effective strain occurring around unclosed foundry voids for the anvils analyzed in the article, divided into the first and second steps.

To determine the influence of the deformation basin shape on the closing of internal foundry voids, the authors also presented the research results in [18].

By analyzing the data presented in Figure 44, Figure 45 and Figure 46, the authors try to answer the question of how the shape of the hitting surfaces of the tools affects the closing of internal foundry voids.

**Figure 44 materials-16-05431-f044:**
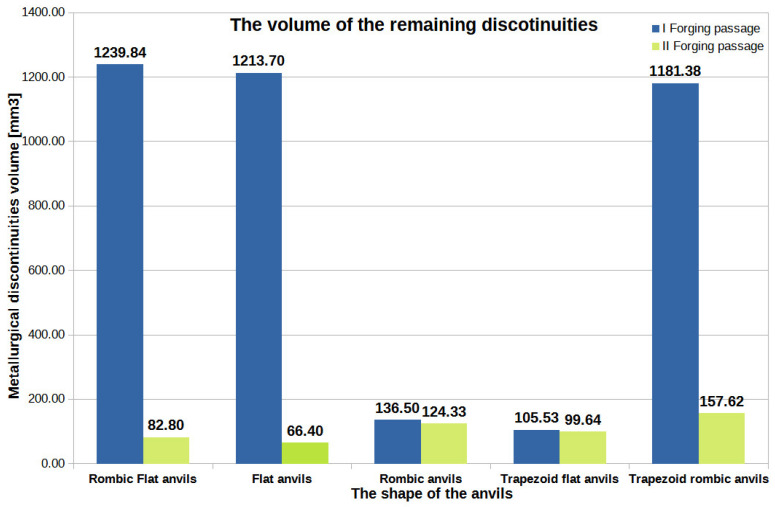
The total volume of unclosed foundry voids in the volume of a forged Zr alloy bar for two forging steps in relation to individual tools’ compositions.

By analyzing the data presented in Figure 44, it can be seen that the best results in terms of the closing of foundry voids during the forging process of Zr alloy bars after the first and second steps were obtained for rhombic–flat and flat tools [18]. After the second step, the values of the total volumes of unclosed foundry voids ranged from 66.40 mm^3^ to 82.80 mm^3^, which constituted only 1.5% to 2% of the volume of the initially modeled discontinuities. It is worth noting that the total starting value of the volume of all modeled foundry voids was 4239 mm^3^. The conducted research shows that the closing of foundry voids can also be performed with the use of rhombic trapezoidal and flat trapezoidal anvils. For these tools, after the first step, an observed reduction in the total volume of foundry voids amounted to 105.53 mm^3^ and 1181.38 mm^3^, respectively. After the second step for these tools, a decrease in the total value of the volume of unclosed foundry voids was noted. The volumes of unclosed foundry voids were 99.64 mm^3^ and 157.62 mm^3^, respectively, which accounted for 2.3% and 3.7% of the volume of the initially modeled foundry voids.

**Figure 45 materials-16-05431-f045:**
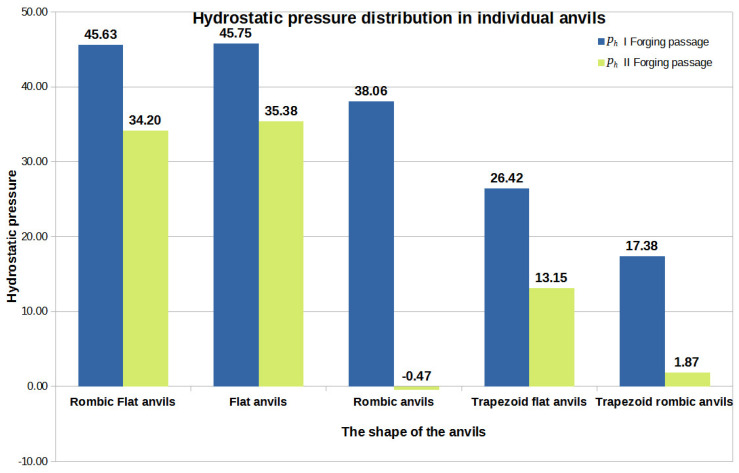
The arithmetic mean value of the hydrostatic pressure around the unclosed foundry voids in relation to the individual tools’ compositions.

Figure 45 also presents data on the value of hydrostatic pressure obtained after forging processeswith flat and shaped tools presented in [18] to compare the hydrostatic pressure values obtained for different anvils. The data presented in Figure 45 show that, for the assembly of flat–trapezoidaland rhombic–trapezoidal anvils, the value of the hydrostatic pressure after the first step was similar and amounted to 26 MPa and 17 MPa, respectively. After the second step, a decrease in this value was observed for both the first and the second anvils’ composition, while for the rhombic–trapezoidal anvils, the decrease was about 16 MPa.

Based on the analysis of the data shown in Figure 45, it can be seen that the highest value of hydrostatic pressure after the first and second steps was obtained for the rhombic–flat and flat tools. The high value of the hydrostatic pressure around the foundry voids is not unambiguous in relation to the total value of the closed foundry voids (Figure 44).

**Figure 46 materials-16-05431-f046:**
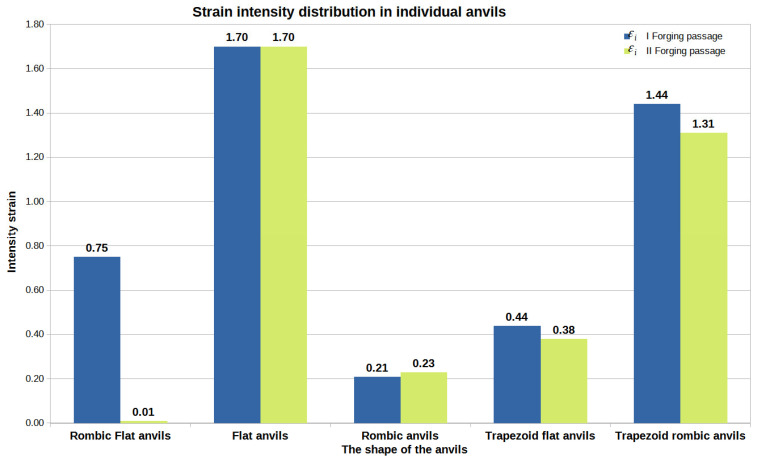
The arithmetic mean value of the deformation intensity occurring around unclosed foundry voids in relation to individual tool compositions.

Figure 46 compares the data on the value of the effective strain obtained after the forging process in flat and shaped tools presented in [18]. The study of the information in Figure 46 shows that the most favorable arithmetic mean value of the effective strain after two steps was obtained for flat tools and was equal to 1.70. The arithmetic mean values of the effective strain calculated for the bar forging in the remaining tool compositions, except for the rhombic–trapezoidal tools, had much lower values. In the implementation of the first two steps with the rhombic–trapezoidal anvils, the arithmetic mean value of the deformation intensity after the first transition was 1.44, and after the second transition, it was 1.31. These values were, however, slightly lower than the values obtained for the deformation with flat anvils. This slightly smaller difference resulted in significant inhibition of the welding process of metallurgical discontinuities after the second forging transition. For trapezoidal anvils, the worst results were obtained in terms of closing of the foundry voids (Figure 44).

Considering the operations of the forging processes of Zr alloy with five different tool compositions, it can be stated that it is favorable, in terms of closed foundry voids, to maintain the effective strain in the area of the closed foundry voids at the level of 1.70 and higher and the hydrostatic pressure at the level of 45 MPa and higher.

## 9. Conclusions

Based on the investigation of the results of the conducted research, the following conclusions were drawn:The use of different hitting surfaces of the tools to close internal foundry voids significantly affects their closing;The most important factor for the closing of internal foundry voids in elongation is not only the value of hydrostatic pressure, which, depending on the anvil’s shape used, should be between 53 and 126 MPa, but, above all, the shape of the deformation basin, which influences the distribution of effective strain;The highest values of the effective strain, in the range of 1.31–1.44, were obtained during the operation of the forging process of zirconium alloy bars with rhombic–trapezoidal tools, providing good material processing and at the same time contributing to obtaining better mechanical properties of the finished product;Closing of all modeled metallurgical discontinuities for both anvil compositions was achieved in the third step;The greatest axial discontinuity was closed with the flat–trapezoidal anvils;When carrying out the elongation operation to weld foundry voids, the highest possible values of relative reduction in the range of 30–40% should be used because high values of reduction affect the formation of stresses and compressive deformations in the local zones of the deformed bar that favor the closing of foundry voids.

After the analysis of the forging process operation to weld the internal foundry voids occurring in Zr alloy ingots, the following guidelines for the forging technology were proposed.

For the initial stages of the elongation operation, anvils with different working surfaces should be used, based on which it is possible to weld the largest (in terms of volume) metallurgical discontinuities in the rod axis. In addition, the use of these anvils results in good material processing and allows for obtaining a homogeneous distribution of mechanical properties in the starting volume of the deformed bar.After the first two steps of the forging operation, additional heating of the bar should be applied.For the final stages of forging, it seems reasonable to use flat tools due to the higher values of the effective strain and hydrostatic pressure in the volume of the deformed bar.In all forging transitions, irrespective of the anvils used, the highest possible values of relative reduction should be used.

The authors intend to verify the obtained modeling results under laboratory conditions in the near future.

## Figures and Tables

**Figure 1 materials-16-05431-f001:**
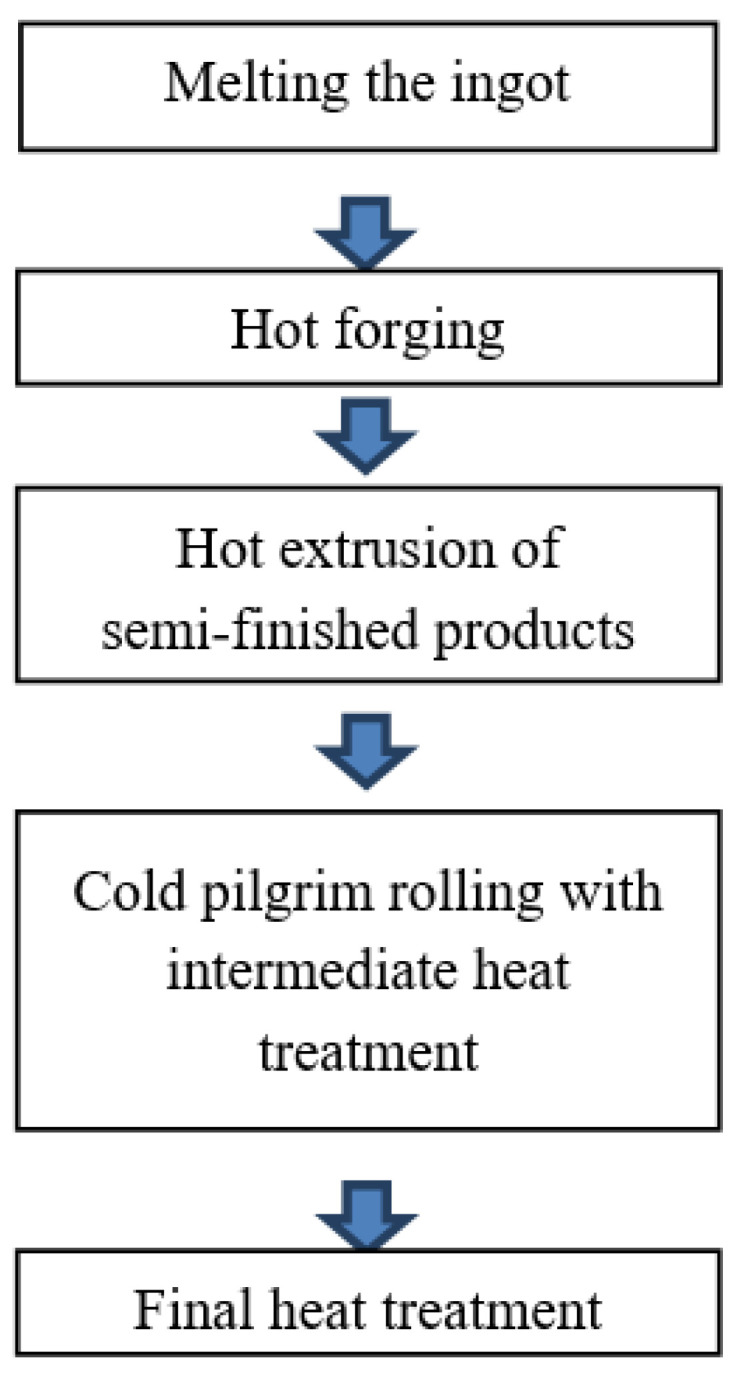
The technology operations in manufacturing tubes and rods from the Zr–1%Nb alloy.

**Figure 2 materials-16-05431-f002:**
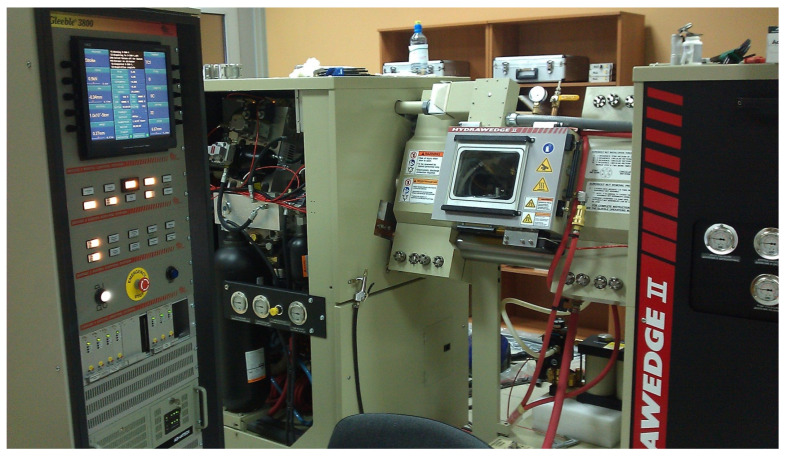
The Gleeble 3800 plastometer testing equipment.

**Figure 3 materials-16-05431-f003:**
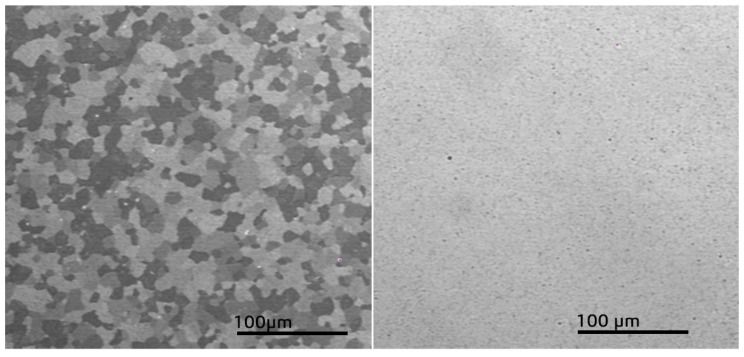
The microstructure of starting specimens of the M5 alloy.

**Figure 4 materials-16-05431-f004:**
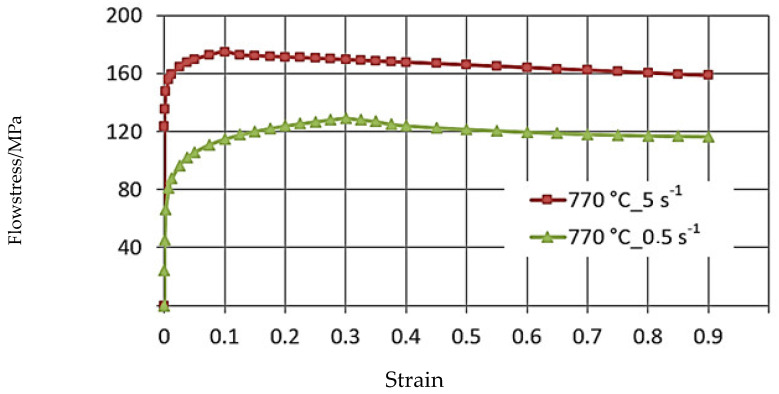
The $\sigma_{P}-\varepsilon$ flow curves of the M5 alloy, obtained for T = 770 °C in the deformation speed range $\dot{\varepsilon }$ from 0.5 to 5.0 s^−1^ using the Gleeble 3800.

**Figure 7 materials-16-05431-f007:**
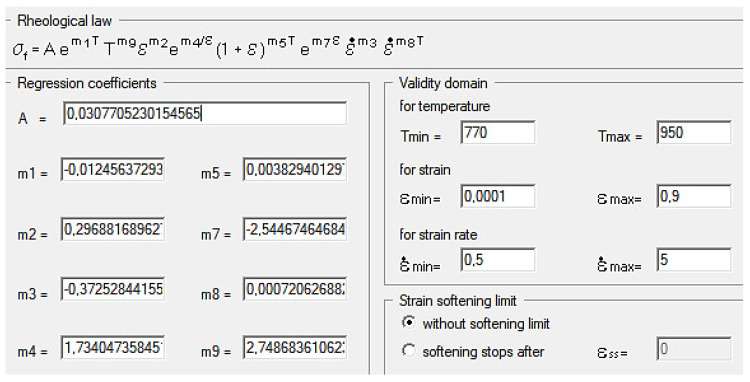
Values of the equation coefficients for the zirconium alloy.

**Figure 8 materials-16-05431-f008:**
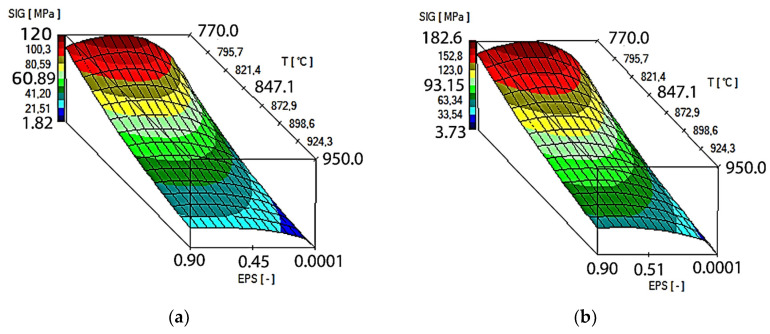
Results of the approximation of the M5 alloy stress–strain curves in the temperature range T = 770–950 °C in the form of three-dimensional graphs: (**a**) $\dot{\varepsilon }$ = 0.5 s^−1^; (**b**) $\dot{\varepsilon }$ = 5.0 s^−1^.

**Figure 9 materials-16-05431-f009:**
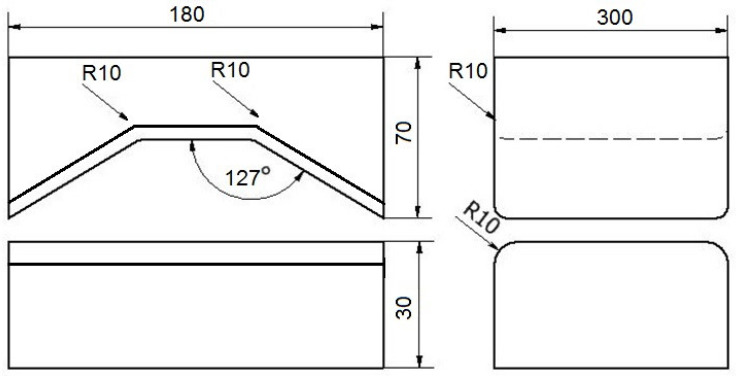
Contour and size of flat trapezoidal tools used for deformation of Zr alloy.

**Figure 10 materials-16-05431-f010:**
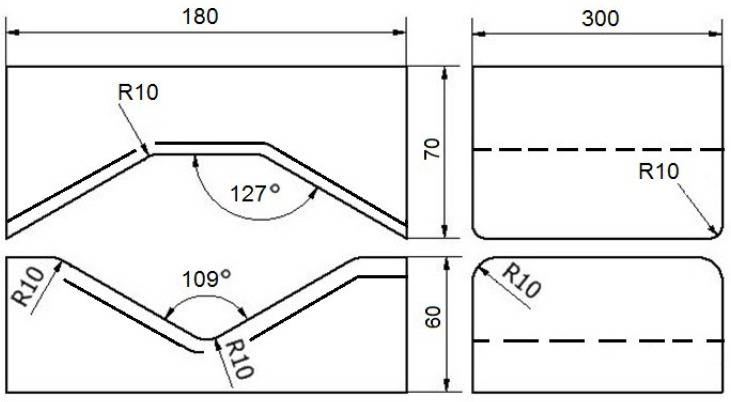
Contour and size of rhombic trapezoidal tools used for deformation of Zr alloy.

**Figure 11 materials-16-05431-f011:**
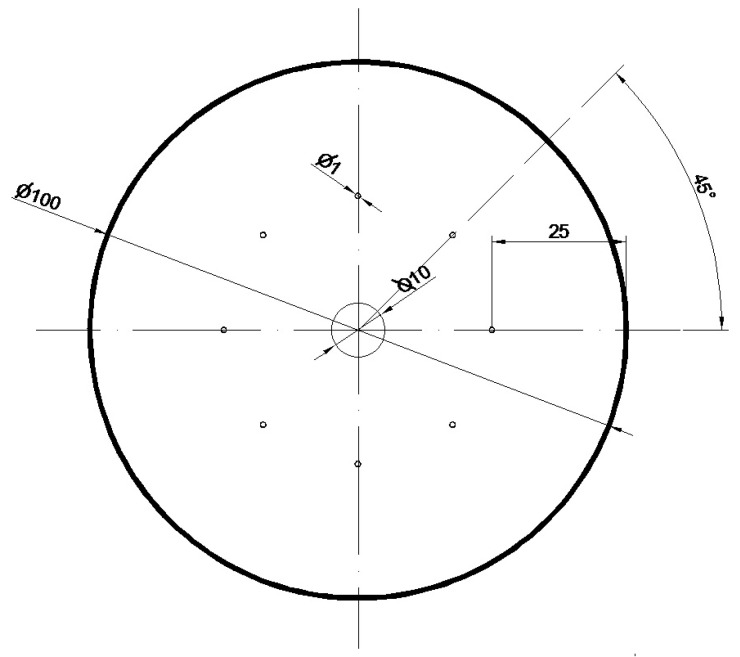
Geometry and distribution of the modeled foundry voids on the end face of the model Zr alloy bar.

**Table 1 materials-16-05431-t001:** Values of the boundary and initial conditions used in the numerical model.

Parameter Name	Parameter Value
Heat transfer coefficient material-anvils	10,000 W/m^2^K
Heat transfer coefficient material-environment	10 W/m^2^K
Relative reduction	35%
Upper anvil speed	8 mm/s
Friction coefficient	0.3
Starting material temperature	950 °C
Starting anvil temperature	250 °C
Environmental temperature	25 °C
Rotary in elongation operation	90 °C

## Data Availability

Not applicable.

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
