# Peer review of "Modeling of Closure of Metallurgical Discontinuities in the Process of Forging Zirconium Alloy"

_materials, 2023, doi:10.3390/ma16155431_

Round 1

Reviewer 1 Report

Dear author(s), please find below suggestions that may justify my final evaluation of the reviewed manuscript ‘Modeling of closure of metallurgical discontinuities in the process of forging zirconium alloy, Manuscript ID: materials-2490002.

Generally, the paper's idea is interesting, and the topic is up-to-date, but it needs improvement to be accepted in reputed journals.

1.     Abstract is lacking key findings of the study.

2.     Remove all typo/Format mistakes, see Lines 24, 113, 106,123, 128, 210-211, 213, 

3.     Lebel the flow operation as a Figure and explain the results in text in lines 46-56.

4.     Line 67 needs more clarification as there are many papers available in literature regarding this issue, even your own published paper in 2023.

5.     The literature review section needs major revision to include more information about the alloys and published literature on the same topic. Include all recent paper on the studied topics.

6.     Lines 93 and 94 need ref.

7.     Equation 1 needs redrafting.

8.     How many runs were used in experimentation of Figures 1-3.

9.     In Figure 2, above 0.2 strain, what causes the softening in studied alloy.

10.  Resolution of Figure 5 is poor.

11.  It can be assumed that the coefficients of the approximating function (1) are well- chosen if the average error does not exceed 15%. A reference is mentioned but can you elaborate further how much errors it will cause in Computational analysis.

12.  Include figure of Gleeble 3800 plastometer testing.

13.  Line 186, Constant parameters of the elongation operation were assumed need more elaboration.

14.  Line 191 sentence is incomplete “A description of the temperature, force, stress, strain and thermomechanical and frictional laws used in the investigation can be found in.”.

15.  Justify the claim of using μ = 0.3?

16.  Are the void sizes defined as per the observed one or they are selected randomly?

17.  Line 271, Figure 41 is missing.

18.  How the voids surface are treated during meshing and elimination of voids in forging process.

19.  Is the mesh convergence were determined for the current study and needed for the readers.

20.  Details of the computational facility used for the current study must be included in the article.

21.  Details of the total number of mesh points and elements must be included in the revised article.

22.  The friction model used in the current analysis need to be included in the FE model section.

23.  How are distorted elements treated in the forging simulation?

24.  Conclusion needs to be more conclusive.

Author Response

Thank you for your comment. Suggested changes have been made to the article. Some questions are answered in the file or below.

Reviewer 2 Report

1.  The author mentioned that the main functionality of this method is the ability to obtain relatively large strain values (up to ε = 1.2), but the true strain is up to 0.9 in Fig. 1. These are contradictory.

2. Please give the microstructure of starting M5 alloy rod.

3. How to count the volume of unclosed foundry voids? 

4. It would be better if the SEM or OM of forged Zr alloys were given to verify the modelling results in the present research.

The English spelling should be polished clearly because there are many errors in this manuscript.

Author Response

Thank you for your comment. Suggested changes have been made to the article. Some questions are answered below.

Reviewer 3 Report

Dear authors,

I am pleased to inform you that the review of the assigned manuscript was conducted.

Generally, the manuscript has several positive features. First of all, the purpose of the work was clearly stated. Secondly, the work was well organized, well discussed, and in most parts, well written. And lastly, there is a wide range of useful results and technical discussions in the manuscript. Accordingly, the manuscript has some important contributions to the advancement of knowledge in the area of forging.

Although the manuscript has high quality, there are some points that should be modified or clarified by the authors.

Reviewer’s comments

Technical comments:

- Section “1. Introduction”: The literature review is weak and needs to be enhanced. The authors are requested to add the following relevant papers to Section “1. Introduction” to improve the quality of the research background:

https://link.springer.com/article/10.1007/s12289-022-01735-y

https://www.sciencedirect.com/science/article/abs/pii/S0924013609003549?via%3Dihub

https://www.sciencedirect.com/science/article/abs/pii/S152661252100788X?via%3Dihub

- Section “2. PURPOSE AND SCOPE OF WORK” should be brought after Section “1. Introduction”.

- Line 189: The authors are requested to show the applied thermal and mechanical boundary conditions in the finite element model.

- Line 195: a thermo-viscoELASTIC model or a thermo-viscoPLASTIC model ?

- Line 198: How was the Coulomb friction coefficient of 0.3 determined? Is it calibrated, assumed, or captured from a paper/Forge’s user manual?

- Section “5. METHODOLOGY OF CONDUCTED NUMERICAL RESEARCH”: The input data in the FE model including mechanical and thermal properties were not given in tabular forms in the manuscript.  

Grammar, vocabulary, and format comments:

The English and syntax of the manuscript should be improved. For example:

- Line 40: The article “the” is redundant in “…for the making..”.

- Line 69: “The authors in” should be replaced with “Banaszek et al.”

- Line 81: “developed” should be replaced with “make”. (Make recommendations)

- Line 112: “illustration” should be replaced with “illustrate”.

- Line 113: “Fig. 1−3 shows” should be replaced with “These figures show”.

- Line 126: “demonstrations” should be replaced with “demonstrates”.

- Line 192: found in ?? (what?).

- Line 205: “has not” should be replaced with “does not”.

- Line 216: “showed in fig. 6 and 7.” should be replaced with “shown in Figs. 6 and 7.”.

- Line 248: “planned” should be replaced with “drawn”.

- Line 305: “Fig.” should be replaced with “Figs.”.

- Line 346: “no hydrostatic pressure not exist” should be replaced with “no hydrostatic pressure exists”.

- Line 468: It is better to show the unit for hydrostatic pressure on Fig. 42.

- Line 510: “not only the value of the hydrostatic pressure but most of all the shape of” should be replaced with “the value of the hydrostatic pressure and mostly the shape of”.

- The maximum values on the vertical axis in Figs. 1-3 should be set on 200 MPa to better see the differences.

- Line 46: It is better to assign a figure to the flowchart showing the operations.

In conclusion, if the authors apply the corrections to the manuscript, the manuscript will be suitable for publication in Materials.

Regards,

The reviewer

July 1, 2023

The English and syntax of the manuscript should be improved.
Please refer to the comments and suggestions.  

Author Response

(The authors gave the same response as above.)

Round 2

Reviewer 1 Report

The literature review section has been improved but needs more work to include all articles on studied alloys and process. 

Author Response

Hello, all comments have been taken into account. Detailed answer in the uploaded file.

Reviewer 2 Report

The revised manuscript can be accepted.

The English language of this article is understandable.

Author Response

(The authors gave the same response as above.)
